# Beyond Marginals: Learning Joint Spatio-Temporal Patterns for Multivariate Anomaly Detection

**Padmaksha Roy**                                                          *padmaksha@vt.edu*
*Department of Electrical and Computer Engineering*
*Virginia Tech*

**Almuatazbellah Boker**                                                     *boker@vt.edu*
*Department of Electrical and Computer Engineering*
*Virginia Tech*

**Lamine Mili**                                                              *lmili@vt.edu*
*Department of Electrical and Computer Engineering*
*Virginia Tech*

**Reviewed on OpenReview:** *https://openreview.net/forum?id=iETTv1okjX*

## Abstract

In this paper, we aim to improve multivariate anomaly detection (AD) by modeling the *time-varying non-linear spatio-temporal correlations* found in multivariate time series data . In multivariate time series data, an anomaly may be indicated by the simultaneous deviation of interrelated time series from their expected collective behavior, even when no individual time series exhibits a clearly abnormal pattern on its own. In many existing approaches, time series variables are assumed to be (conditionally) independent, which oversimplifies real-world interactions. Our approach addresses this by modeling joint dependencies in the latent space and decoupling the modeling of *marginal distributions, temporal dynamics, and inter-variable dependencies*. We use a transformer encoder to capture temporal patterns, and to model spatial (inter-variable) dependencies, we fit a multi-variate likelihood and a copula. The temporal and the spatial components are trained jointly in a latent space using a self-supervised contrastive learning objective to learn meaningful feature representations to separate normal and anomaly samples.

## 1 Introduction

Modern industrial systems rely on networks of interconnected sensors that produce vast streams of multivariate time series data during operation. Detecting anomalies in these data plays a critical role in identifying faults early, mitigating security threats, and maintaining system reliability and safety. In industrial settings, time-series data are frequently used to monitor the performance of machines, IT infrastructure, spacecraft, and engines. Anomaly detection has become a vital component of time series analysis, enabling early detection of faults and preventing potential failures. Recent advances in deep learning have spurred the development of various methods to address this problem. For instance, recurrent neural networks (RNNs) Hundman et al. (2018); Su et al. (2019); Canizo et al. (2019) have been widely used to capture temporal dependencies in multivariate sequences. Meanwhile, other approaches employ graph-based models or Transformer architectures Vaswani (2017) to focus on variable relationships and sequential patterns Deng & Hooi (2021), Anomaly Transformer Xu (2021). These models effectively utilize temporal structures and adapt neural networks to time series tasks. Despite these advancements, the detection of anomalies in multivariate time series data remains a challenge. The primary difficulty arises from the intricate temporal dependencies and correlations between multiple variables. Anomalies often manifest as subtle deviations that are hard to

isolate from natural fluctuations without contextual awareness. In addition, real-world datasets frequently suffer from noise, missing values, and high dimensionality, further complicating the modeling process. A major limitation in this domain is the scarcity of labeled data. In many cases, it is unclear during training whether a given point represents an anomaly. This lack of ground-truth labels has driven the adoption of unsupervised learning approaches. Methods such as autoencoders and adversarial networks attempt to model data distributions without labels to identify deviations. However, unsupervised approaches often struggle with contextual anomalies and dependencies between variables, making detection unreliable in complex scenarios.

Multivariate time-series data, especially high-order multivariate time series (HO-MTS), introduce additional layers of complexity that make anomaly detection particularly challenging. Unlike univariate time series, which involve a single variable observed over time, HO-MTS captures interdependencies between multiple variables, not just at a single time step but also across multiple time lags. This temporal and cross-variable dependency structure amplifies the difficulty of modeling and detecting anomalies. HO-MTS data require models to account for both spatial correlations (relationships between variables) and temporal dependencies (relationships across time steps). For instance, a sensor measuring pressure at time $t$ might depend on the temperature reading at time $t-1$ or flow rate at time $t-2$. These dependencies often span long time horizons, making it necessary to handle lagged interactions effectively. Traditional models, such as autoregressive methods, struggle to capture such intricate relationships, particularly when the data have nonlinear patterns or dependencies that are not explicitly observable. This challenge is compounded when anomalies arise from unexpected combinations of variable interactions rather than simple threshold violations, requiring models to analyze contextual anomalies instead of point anomalies. In HO-MTS, anomalies can occur as contextual deviations rather than isolated outliers. For example, a sudden spike in temperature may not be anomalous if it follows an increase in pressure, but it could indicate a fault if the pressure remains constant. Anomalies are assumed to diverge from these correlated patterns of changes among multiple variables, considering their joint distributions.

Supervised learning methods, on the other hand, present an attractive alternative when labeled data is available. These approaches can explicitly learn patterns associated with anomalies and differentiate them from normal behavior, especially when anomalies are subtle or involve relationships between multiple variables. Industrial datasets often contain very few labeled anomalies. Recent techniques, such as semi-supervised learning and contrastive learning, enable models to utilize a small set of labeled data while benefiting from larger unlabeled datasets. Supervised techniques also benefit from the ability to incorporate domain knowledge through labeled examples, improving interpretability and performance. Pre-trained models and data augmentation techniques can leverage small amounts of labeled data to improve performance significantly. Although unsupervised methods are useful when labeled data is unavailable, supervised learning provides better performance in scenarios where labeled data can be obtained (even in small quantities). It can model complex dependencies, improve interpretability, and leverage domain-specific insights. Detecting anomalies in HO-MTS data requires capturing both temporal dependencies and spatial (variable) correlations effectively. While deep learning models such as Long Short-Term Memory networks (LSTMs) and Transformers have shown success in modeling temporal patterns, they often struggle to adequately capture latent dependencies across variables, especially in scenarios with complex interactions and nonlinear relationships among the variables Tian et al. (2023); Chen et al. (2023); Zheng et al. (2023).

Transformers Vaswani (2017) rely on self-attention mechanisms to capture long-range dependencies in sequences. Recent studies such as Jeong et al. (2023), Tuli et al. (2022), and Xu (2021) underscore the limitations of modern Transformer architectures in jointly capturing spatial and temporal dependencies for effective anomaly detection in multivariate time series data. AnomalyBERT Jeong et al. (2023) introduces a specialized Transformer architecture that incorporates 1D relative position bias to capture these multivariate spatial interactions better. TranAD Tuli et al. (2022) highlight that plain Transformer architectures struggle with detecting subtle anomalies in multivariate data, often failing when deviations are minor. To overcome this, the authors enhance Transformers using adversarial training and self-conditioning to amplify anomaly signals and improve generalization. By incorporating model-agnostic meta-learning, TranAD outperforms basic Transformer models by a huge margin. The authors of Anomaly Transformer Xu (2021) argue that "pointwise" reconstruction or prediction errors treat each timestamp in isolation, so they fail to

capture longer-range dynamics and are easily overwhelmed by the vast majority of normal points. To address this, they introduce two complementary attention maps—*series-association* from the raw self-attention and a learnable *prior-association* that biases toward local context—and define their divergence or association discrepancy as an anomaly score.

Our central contribution is a unified spatio-temporal modeling framework that combines Transformers (for temporal context) with modeling of multivariate joint spatial dependencies—an integration we believe to be novel in the multivariate anomaly detection(AD) literature. We use the original Transformer model Vaswani (2017) solely as a temporal encoder. Rather than modifying the Transformer to capture spatial dependencies, our key innovation lies in integrating it with a theoretically grounded multivariate modeling framework—leveraging Gaussian, Student-t likelihoods, and copula theory—to explicitly model nonlinear spatial dependencies. Following AnomalyBERT Jeong et al. (2023), we assume a small pool of real anomaly representations and then generate additional anomaly sample representations via simple augmentations to ensure that our synthetic anomaly samples remain realistic. Salinas et al. (2019) noted that modeling correlations among variables is critical in multivariate AD and methods that assume independence across time series fail when inter-variable correlations are critical. In anomaly detection, subtle but coordinated deviations across nodes can signal issues even if individual nodes appear normal. While prior works have acknowledged this need, to the best of our knowledge, ours is the first to bring together theoretical and architectural components in a cohesive framework with a simple self-supervised contrastive learning objective. Our contribution can be summarized as follows:

- We introduce an end-to-end training framework that jointly optimizes a Transformer encoder to extract temporal information in high-dimensional time series and a multivariate likelihood or Copula model to capture the latent space spatial dependency. By backpropagating through both components simultaneously, the model discovers latent embeddings that preserve local and long-range time relations while also conforming to consistent variable-to-variable dependencies.

- Instead of labeling individual samples, we treat each window or frame as a coherent sequence, then build a contrastive loss that enforces normal frames to achieve a high log-likelihood density, while anomalous frames are pushed below a margin in log-likelihood space. This approach better captures short-range temporal structure and ensures that anomalies which exhibit subtle multivariate deviations are effectively separated in the latent embedding space.

- Experiments conducted on multiple public benchmark multivariate and synthetic time series datasets demonstrate that both the variants of our model consistently outperform state-of-the-art techniques, achieving higher precision, recall, and AUC-ROC scores, especially when the time series exhibits joint spatio-temporal dependencies.

## 1.1 Related Work

Time series anomaly detection has been approached using both supervised Jia et al. (2019),Cook et al. (2019) and unsupervised methods Audibert et al. (2020), Zhang et al. (2021), Thill et al. (2021), each catering to different challenges posed by the data. Unsupervised methods dominate this field due to the lack of labeled anomalies in most real-world datasets. Techniques such as autoencoders, isolation forests, and Gaussian mixtures focus on modeling the distribution of normal data and identifying deviations as anomalies. Deep learning models, such as LSTMs and Autoencoders, leverage reconstruction errors or forecasting residuals to detect anomalies without requiring labels. However, these methods often struggle with contextual and correlated anomalies, especially in multivariate settings where relationships between variables evolve over time. Supervised methods, on the other hand, utilize labeled datasets to explicitly distinguish anomalies from normal patterns. Approaches such as RNN classifiers, attention-based networks Wang & Liu (2024); Zhao et al. (2020), and graph neural networks Zhao et al. (2020), Deng & Hooi (2021) excel at learning complex dependencies and classifying anomalies when labeled data is available. Recent advances in semi-supervised learning Akcay et al. (2019) and transfer learning have further extended supervised approaches to scenarios with limited labeled data, offering improved accuracy and interoperability. While unsupervised methods are widely used due to their flexibility, supervised approaches are gaining traction as they can better

capture latent dependencies and nonlinear correlations in high-order multivariate time series, particularly when integrated with copula-based models to enhance dependency modeling in latent spaces. Previously, encoder-decoder based architectures integrated with adversarial training framework Audibert et al. (2020) that leveraged the strengths of both autoencoders and adversarial training while addressing the shortcomings of each approach were developed for multivariate time series anomaly detection.

TranAD Tuli et al. (2022) leverages attention-based encoders, self-conditioning, adversarial training, and MAML for robust, efficient, and data-efficient anomaly detection and diagnosis. STADN Tian et al. (2023) integrates spatial-temporal information using graph attention and LSTM networks, predicts sensor behavior, and enhances anomaly detection by reconstructing prediction errors for better discrimination. Anomaly-BERTJeong et al. (2023) addresses this issue by introducing a data degradation scheme for self-supervised model training. They specifically define four types of synthetic outliers and propose a degradation process in which parts of the input data are replaced with these outliers. In addition to leveraging the self-attention mechanism of the Transformer architecture, their approach transforms multivariate data points into temporal representations enriched with relative position bias and computes anomaly scores based on these representations. TACTis Drouin et al. (2022) addresses the challenge of estimating the joint predictive distribution for high-dimensional multivariate time series by introducing a flexible approach built on the transformer architecture, leveraging an attention-based decoder that is theoretically proven to replicate the behavior of non-parametric copulas. Anomaly-Transformer Xu (2021) identifies anomalies by leveraging their tendency to form concentrated associations with adjacent points, unlike normal data which associates more broadly across the series. CARLA Darban et al. (2025) leverages contrastive learning and a self-supervised strategy to enhance anomaly detection by learning similar representations for adjacent windows and distinguishing anomalies based on proximity to the representation. Another recent work Tang et al. (2024) addressed the problem of time series anomaly detection with self-supervised contrastive learning by designing a perturbation classifier to infer the pseudo-labels of data perturbations.

## 2 Proposed Framework

In this section, we elaborate the deep learning framework, contrastive loss function and the training strategy that we considered to develop our model.

### 2.1 Problem Statement

In our problem, we consider a multivariate time series $\mathbf{X} \in \mathbb{R}^{D \times T}$ arising from a cyber-physical system, where $D$ is the dimension of each time frame and $T$ is the total length of the time series. We embed $\mathbf{X}$ into an embedding space $\mathbf{Z} \in \mathbb{R}^{d \times T}$, capturing the most influential features. Our goal is to design a contrastive loss that leverages the *joint distribution* (or *mutual information*) among these latent variables, as measured via a *copula log-density* or a multivariate likelihood method, to effectively *separate* normal from anomalous frames. Intuitively, normal data lies in a *higher log-likelihood of the density function* region in the latent space, reflecting consistent dependency patterns, whereas anomalies disrupt these dependencies and thus spread over a *lower log-likelihood of the the density function*. By constructing a contrastive objective that enforces a large gap in the log-density (or mutual information) between normal and anomalous frames, we exploit the fundamental distinction in their dependence structures to enhance *anomaly detection* within the latent feature space.

In this approach, each short subsequence of length $L$ in the multivariate time series is labeled as a *frame*, which can be either normal (label 0) or anomalous (label 1). *(A) Labeling Frames.* If any timestamp in a length-$L$ snippet is anomalous, then the entire frame is labeled as anomaly; otherwise, it is normal. This ensures that anomalies spanning multiple time steps are not overlooked. The anomalies are introduced at random positions, and their proportions vary from 25% to 50% of the normal window length.

*(B) Generating Frame Pairs.* Once frames are labeled, we form pairs $(\mathbf{x}_i, \mathbf{x}_j)$ that may be normal–normal, normal–anomaly, or anomaly–anomaly. Such pairs can then be used in contrastive or mutual-information-based learning, since each pair directly encodes local temporal dependencies and interactions among variables. If either sequence is anomalous, we label the pair as anomaly-involved.

*(C) Advantages for Time-Series.* By treating small windows as frames rather than single samples, we can better capture short-range temporal correlations, especially when anomalies or key dependency patterns span multiple points.

*(D) Estimating the multi-variate log-density.* Computing mutual information or fitting a copula or a multivariate likelihood requires a sufficient number of samples and a joint view of multiple dimensions; having short subsequences provides richer data for these estimations. This approach results in more stable estimates of dependency patterns, allowing the model to differentiate high-log-density regions (representing normal data) from low-log-density regions (representing anomalies).

## 2.2 Transformer Encoder

In this setting, we employ a Transformer encoder to project: $f_\theta : \mathbb{R}^{D \times T} \to \mathbb{R}^{d \times T}, \quad \mathbf{X} \mapsto \mathbf{Z}$. A Transformer processes the entire time series in parallel by applying self-attention across all time steps. This mechanism allows every position in the sequence to reach every other position, capturing both local and long-range dependencies. In particular, for longer time horizons, the model is less prone to forgetting distant signals compared to recurrent networks. Hence, each latent vector $\mathbf{z}_t$ (corresponding to a snippet or frame in time) encodes the key temporal patterns that matter to distinguish normal from anomaly sequences. Each dimension of $\mathbf{z}$ emerges from attention-based feature extraction, so normal data cluster around consistent patterns, while anomalies, which break typical variable interactions, are projected elsewhere in the latent space. This dimensionality reduction clarifies which features (and which timesteps) are most crucial to anomaly detection. We use the standard encoder based on the paper Vaswani (2017). We utilize the Transformer encoder layer from the PyTorch library, which comprises a self-attention mechanism and a feedforward network. Unlike Jeong et al. (2023), we did not implement their custom relative positional encoding, as our focus is solely on employing the Transformer model as an encoding network.

## 2.3 Dependency Modeling in Latent Space

In our anomaly detection framework, the latent representation $\mathbf{z} \in \mathbb{R}^d$ captured by a Transformer encoder undergoes a joint dependency modeling among the dimensions of $\mathbf{z}$. In order to learn the correlation structure, we mainly focus on estimating the log-density of a multivariate gaussian and student-t likelihood (elaborated in Algo 1), and the copula density of two main copula families: the *Gaussian copula* and the *Student-t copula* (detailed in Algo 2). Each of these approaches primarily learns a *correlation structure* in the latent space, allowing us to evaluate how well a given $\mathbf{z}$ aligns with typical (normal) behavior.

### 2.3.1 Dependency Modeling with Copulas

A copula models how the embedding dimensions $(z_1, \ldots, z_d)$ co-vary, focusing on their dependency structure irrespective of individual marginal distributions. If normal embeddings $\mathbf{z}$ maintain certain correlational patterns (e.g., $z_1$ rises when $z_3$ drops), the copula assigns them high log-density. Anomalies produce embeddings that violate these learned dependencies, thus yielding lower copula log-density.

Sklar's Theorem states that if $F$ be any $d$-variate CDF with continuous marginals $F_1, \ldots, F_d$, then there exists a unique copula $C$ such that

$$F(z_1, \ldots, z_d) = C\big(F_1(z_1), \ldots, F_d(z_d)\big).$$

Conversely, if $C$ is a copula and $F_1, \ldots, F_d$ are univariate CDFs, then the right-hand side defines a valid joint CDF.

The *Gaussian copula* with correlation matrix $\Sigma$ is defined by:

$$C_\Sigma(u_1, \ldots, u_d) = \Phi_\Sigma\big(\Phi^{-1}(u_1), \ldots, \Phi^{-1}(u_d)\big),$$

where $\Phi^{-1}$ is the inverse CDF of a standard normal, and $\Phi_\Sigma$ is the $d$-variate standard normal CDF with correlation $\Sigma$.

The *Student-t copula* with correlation matrix $\Sigma$ and $\nu$ degrees of freedom is:

$$C_{t,\Sigma,\nu}(u_1,\ldots,u_d) = T_{\Sigma,\nu}\big(T_\nu^{-1}(u_1),\ldots,T_\nu^{-1}(u_d)\big),$$

where $T_\nu^{-1}$ is the inverse CDF of the univariate Student-$t(\nu)$, and $T_{\Sigma,\nu}$ is the multivariate Student-$t$ CDF.

---

**Algorithm 1** Dependency Modeling via Log-Density of Multivariate Likelihood

---

**Input**   : Latent vector $\mathbf{z} \in \mathbb{R}^d$, Cholesky parameters $\phi$, dof $\nu$ (for Student–$t$)
**Output:** Log–likelihood $\log c(\mathbf{z}; \phi, \nu)$

1. Standardise the latents:
$$\mathbf{z}_{\text{std}} \leftarrow \frac{\mathbf{z} - \mu}{\sigma}.$$

2. Transform unknown marginals to gaussian/student-t marginals(0,1):
$$u_i \leftarrow \begin{cases} \Phi\big(z_{\text{std},i}\big), & \text{Gaussian,} \\ T_\nu\big(z_{\text{std},i}\big), & \text{Student–}t, \end{cases} \quad i = 1,\ldots,d.$$

3. Quantile transform:
$$z_i^{(\text{transf})} \leftarrow \begin{cases} \Phi^{-1}(u_i), & \text{Gaussian,} \\ T_\nu^{-1}(u_i), & \text{Student–}t, \end{cases} \quad i = 1,\ldots,d.$$

4. Correlation matrix: unpack $\phi \to L$ (lower-triangular, diag$\to$softplus); $\Sigma \leftarrow L L^\top$. 5. Log–likelihood:

$$\log c(\mathbf{z}) = \begin{cases} -\frac{1}{2}\Big(z^{(\text{transf})^\top}\Sigma^{-1}z^{(\text{transf})} + \log\det\Sigma + d\log 2\pi\Big), & \text{Gaussian,} \\ \log\Gamma\big(\frac{\nu+d}{2}\big) - \log\Gamma\big(\frac{\nu}{2}\big) - \frac{1}{2}\log\det\big(\nu\pi\,\Sigma\big) \\ \quad -\frac{\nu+d}{2}\log\Big(1 + \frac{1}{\nu}\,z^{(\text{transf})^\top}\Sigma^{-1}z^{(\text{transf})}\Big), & \text{Student–}t. \end{cases}$$

**return** $\log c(\mathbf{z}; \phi, \nu)$

---

---

**Algorithm 2** Dependence Modeling via True Copula Log–Density

---

**Input**   : Latent vector $\mathbf{z} \in \mathbb{R}^d$, marginal CDFs $\{F_i\}_{i=1}^d$, Cholesky parameters $\phi$, degrees of freedom $\nu$ (for Student–$t$)
**Output:** Log–copula density $\log c(\mathbf{u}; \Sigma, \nu)$

1. Standardise the latents to zero mean and unit variance: $\mathbf{z}_{\text{std}} \leftarrow (\mathbf{z} - \mu)/\sigma$.
2. Transform marginals using probability integral transform $u_i \leftarrow F_i(z_{\text{std},i}) \in [0,1]$, $i = 1,\ldots,d$.
3. Impose standard Normal/Student-t margins by quantile transform

$$v_i \leftarrow \begin{cases} \Phi^{-1}(u_i), & \text{Gaussian,} \\ T_\nu^{-1}(u_i), & \text{Student–}t, \end{cases} \quad i = 1,\ldots,d.$$

4. Estimate dependence via the correlation matrix: unpack $\phi \to L$ (lower-triangular, diag$\to$ softplus); $\Sigma \leftarrow L L^\top$.
5. Compute copula log-density:

$$\log c(u) = \begin{cases} \log\varphi_d(v; \Sigma) - \sum_{i=1}^d \log\varphi_1(v_i), & \text{Gaussian,} \\ \log f_{t_d,\nu}(v; \Sigma) - \sum_{i=1}^d \log f_{t_1,\nu}(v_i), & \text{Student–}t. \end{cases}$$

where $\varphi_d$ and $\varphi_1$, $f_{t_d,\nu}$ and $f_{t_1,\nu}$ are the $d$- and univariate Normal and student-t PDFs
**return** $\log c(u)$

---

Since we concentrate on the *correlation parameters* $\phi$ (which define $L$), our optimization updates only those elements to best fit normal data. Anomalies are detected if they yield low log-likelihood under this Gaussian or student-t correlation structure. The Gaussian copula is simple and captures linear correlation well. It is suitable when tail dependence is not extreme, or when anomalies primarily break moderate cross-variable correlations. More details can be found in our appendix section.

The Transformer encoder ensures each embedding or latent vector $\mathbf{z}$ is an expressive representation of temporal and cross-feature relationships of the original time series. Meanwhile, the copula provides a precise measure of *joint* likelihood for those latent coordinates. Consequently, normal data reside in a dense, high-likelihood region of embedding space, whereas anomaly embeddings occupy a lower-likelihood, lower-density region.

### 2.3.2 Student-$t$ Copula and Student-t Likelihood for Heavy-tailed Dependencies

To address more pronounced tail behavior, we can replace the Gaussian assumption with a **Student-$t$ copula** or a Student-t multivariate likelihood, which has an extra *degrees of freedom* ($\nu$) parameter governing tail thickness.

This approach again optimizes primarily the *correlation parameters* $\phi$ (related to $L$) *and* we may also learn or fix the $\nu$ parameter. Hence, anomalies that yield *unexpected* tail dependencies are flagged with low likelihood. Real-world signals often exhibit outliers or heavy-tailed distributions. A Student-$t$ copula and a Student-t multivariate likelihood accommodates such extremes more naturally than the Gaussian copula, assigning higher probability mass in the tails. Thus, anomalies deviating in a heavy-tailed manner are more cleanly separated. If data indeed show large spikes, a $t$-copula or a student-t multivariate likelihood typically yields more robust anomaly detection than its Gaussian counterpart.

### 2.3.3 Learning the Correlation Parameter $\phi$

In all the cases, our primary focus is on *learning the correlation structure* that defines how the latent dimensions co-vary for normal samples. Concretely, we store a vector of Cholesky parameters $\phi$ and the parameter $\nu$ in the Student-$t$ case, and backprop through these when fitting normal data in training. At inference, any $\mathbf{z}$ that fails to match this learned dependency pattern receives a lower log-likelihood and is deemed more likely anomalous. Throughout, we primarily tune the correlation parameters $\phi$, enabling an end-to-end training scheme where the latent encoder (e.g. a Transformer) supplies embedded features, and the copula measures how well they align with the normative correlation structure.

## 2.4 Contrastive Loss

We model the joint distribution of latent variables $\mathbf{z} = (z_1, \ldots, z_d)$, represented by $c_\phi(\mathbf{z})$, which effectively captures their underlying dependencies irrespective of marginal distributions in case of a copula, and a multivariate *log-density* $\log c_\phi(\mathbf{z})$ can be interpreted as reflecting the *mutual information* among these variables. If $\mathbf{z}$ aligns with the normal dependency structure, $c_\phi(\mathbf{z})$ is high (i.e., strong correlations), whereas anomalies that break these dependencies yield a low copula log-density. Indeed, the copula log-likelihood is the sum of $\log c_\phi$ over samples; thus, maximizing copula likelihood is equivalent to maximizing the multi-dimensional dependency, or high MI, in the embedding space. To exploit this for anomaly detection, we construct a *contrastive loss* that (1) maximizes $\log c_\phi\big(\mathbf{z}_n(\theta)\big)$ for normal samples $\mathbf{z}_n$, $\theta$ are the encoder parameters. The goal is to ensure that normal data inhabit a high-MI region, and (2) forces anomalous samples $\mathbf{z}_a$ into a lower-density region. Specifically, we jointly optimize the *encoder parameters* $\theta$ (mapping raw time series to embedding space $\mathbf{z}$) and *latent space dependency parameters* $\phi$ by minimizing the loss function

$$
\begin{aligned}
\mathcal{L}(\theta, \phi) = -\sum_{n \in \text{Norm}} \log c_\phi\big(\mathbf{z}_n(\theta)\big) + \\
\alpha \sum_{a \in \text{Anom}} \max\{0, \log c_\phi(\mathbf{z}_a(\theta)) - (\mu_{\text{norm}} - \delta)\},
\end{aligned}
\tag{1}
$$

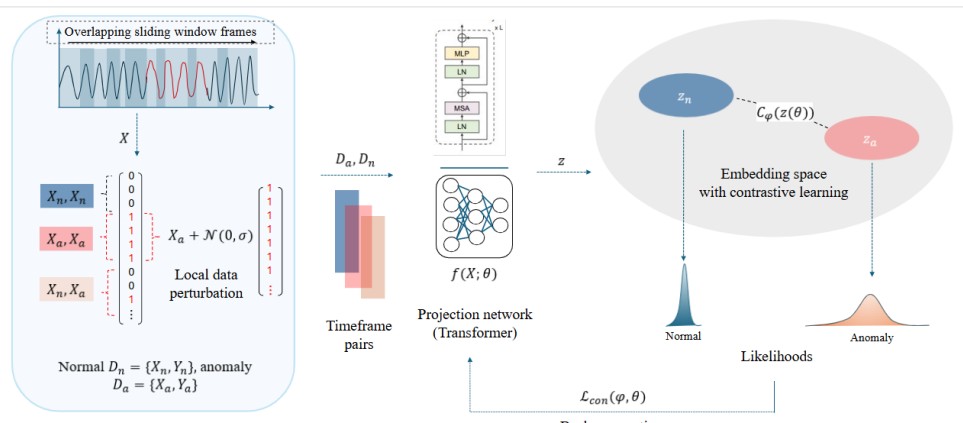

Figure 1: Spatio-Temporal Dependency Aware Feature Learning Framework

where Norm and Anom are normal and anomaly data frames.

A common concern arises because the multi-variate log-density, $\log c_\phi(\mathbf{z}(\theta))$, is often negative for high-dimensional data, so *maximizing* this quantity corresponds to *minimizing* its negative. Hence, for normal samples we add the term $-\log c_\phi(\mathbf{z}_n(\theta))$ in the loss, ensuring that we push those log-densities *toward zero* (i.e., less negative). To separate anomalies, we place a margin constraint that the anomalous log-density stays below $\mu_{\text{norm}} - \delta$, where $\mu_{\text{norm}} < 0$ is the average log-density over normal data and $\delta > 0$ is a margin. We then penalize any anomaly whose log-density $\log c_\phi(\mathbf{z}_a(\theta))$ exceeds $(\mu_{\text{norm}} - \delta)$. Backpropagating with respect to both encoder $(\theta)$ and spatial dependency $(\phi)$ parameters end-to-end guides the model to learn the embeddings that preserve key time-dependent correlations for normal data (thereby raising their mutual information), while anomaly embeddings deviate and incur a higher penalty.

## 2.5 Synthetic Data Generation Scheme

Since our training data have only normal data, we generate degraded inputs by replacing parts of a window with outliers during the training phase. Similarly to Anomaly-BERT Jeong et al. (2023), we randomly select an interval $[t_0, t_1]$ within a window $\mathbf{X} = x_{t_0:t_1}$. The selected sequence $\mathbf{X} = x_{t_0:t_1}$ is then replaced by one of the synthetic outliers described in the following.

### 2.5.1 Local Perturbation

We extend the degradation approach of Jeong et al. (2023) by introducing a local perturbation scheme that synthesizes anomalies through small, controlled modifications of real anomaly snippets. A randomly selected snippet is resized to match the target window and perturbed by adding Gaussian noise, applying slight feature scaling (e.g., $[0.95, 1.05]$), or permuting a few rows to induce temporal variation. The final perturbed snippet is then overlaid onto the target sequence. This approach ensures that the generated anomalies remain close to the real anomaly distribution while introducing variability for robustness. For the local perturbation approach, we utilize 10-15% of the labeled anomalies to create new anomalies that closely mimic the characteristics of true anomalies.

## 3 Experiments and Results

In this section, we begin by outlining the experimental setup. Following that, we conduct a series of experiments to evaluate the effectiveness of the model, classification results, and the results of ablation studies.

## 3.1 Datasets and Baseline Methods

We present experimental results on five widely-used benchmark datasets: SWaT, WADI, SMAP, MSL, and SMD Goh et al. (2017), Ahmed et al. (2017), Hundman et al. (2018), Su et al. (2019). These datasets are derived from various sources, including sensors in server machines, spacecraft, and water treatment or distribution systems. We use a small portion of the labeled anomalies (10-15 %) approximately to generate synthetic anomalies with our local perturbation scheme. Again, we use only the continuous features in all the datasets for modeling. Several advanced models have been proposed for anomaly detection in multivariate time series. MERLIN Nakamura et al. (2020) is a self-supervised method that generates pseudo-labels by learning representations and applies contrastive learning to detect anomalies effectively. LSTM-NDT Hundman et al. (2018) leverages LSTM networks for neural density estimation, capturing temporal dependencies in multivariate time series. DAGMM Zong et al. (2018) combines dimensionality reduction with Gaussian mixture models through a deep autoencoding framework to estimate density and identify anomalies. OmniAnomaly Su et al. (2019) employs a variational RNN-based architecture to model temporal dependencies and reconstruct inputs using stochastic latent variables for anomaly detection. MSCRED Zhang et al. (2019) reconstructs multi-scale signature matrices via a convolutional recurrent encoder-decoder to capture temporal correlations and detect anomalies. MAD-GAN Li et al. (2019) applies an adversarial framework using GANs to reconstruct normal patterns, flagging significant reconstruction deviations as anomalies. USAD Audibert et al. (2020) integrates adversarial training with autoencoders in a unified framework to learn patterns and identify anomalies. MTAD-GAT Zhao et al. (2020) utilizes graph attention networks to effectively capture spatial and temporal dependencies in multivariate time series. CAE-M Zhang et al. (2021) applies a convolutional autoencoder to reconstruct and predict temporal dependencies, facilitating anomaly detection. GDN Deng & Hooi (2021) leverages graph neural networks to model inter-variable dependencies, enhancing anomaly detection performance. Finally, TranAD Tuli et al. (2022) adopts a transformer-based architecture with attention mechanisms to model long-term dependencies and effectively identify anomalies in multivariate time series. We also compared CARLA Darban et al. (2025) and Anomaly-Bert Jeong et al. (2023) as baseline models. The Plain Transformer is the standard architecture from Vaswani et al., with an additional variant incorporating the relative positional bias proposed in AnomalyBERT to capture spatial relationships.

Table 1: Performance metrics (Precision (P), Recall (R), AUC, F1) for various methods across datasets- Means ± std, NA (Not Available) indicates metrics that were not reported in the respective original papers.

| Method | WADI | | | | SWaT | | | | MSL | | | |
|---|---|---|---|---|---|---|---|---|---|---|---|---|
| | P | R | AUC | F1 | P | R | AUC | F1 | P | R | AUC | F1 |
| MERLIN | 0.0636 ± 0.00 | 0.7669 ± 0.04 | 0.5912 ± 0.03 | 0.1174 ± 0.01 | 0.6560 ± 0.03 | 0.2547 ± 0.01 | 0.6175 ± 0.03 | 0.3669 ± 0.02 | 0.2613 ± 0.01 | 0.4645 ± 0.02 | 0.6281 ± 0.03 | 0.3345 ± 0.02 |
| LSTM-NDT | 0.0138 ± 0.00 | 0.7823 ± 0.04 | 0.6721 ± 0.03 | 0.0271 ± 0.00 | 0.7778 ± 0.04 | 0.5109 ± 0.03 | 0.7140 ± 0.04 | 0.6167 ± 0.03 | 0.6288 ± 0.03 | 1.0000 ± 0.05 | 0.9532 ± 0.05 | 0.7721 ± 0.04 |
| DAGMM | 0.0760 ± 0.00 | 0.9981 ± 0.05 | 0.8563 ± 0.04 | 0.1412 ± 0.01 | 0.9933 ± 0.05 | 0.6879 ± 0.03 | 0.8436 ± 0.04 | 0.8128 ± 0.04 | 0.7363 ± 0.04 | 1.0000 ± 0.05 | 0.9716 ± 0.05 | 0.8482 ± 0.04 |
| OmniAnomaly | 0.3158 ± 0.02 | 0.6541 ± 0.03 | 0.8198 ± 0.04 | 0.4260 ± 0.02 | 0.9782 ± 0.05 | 0.6957 ± 0.03 | 0.8467 ± 0.04 | 0.8131 ± 0.04 | 0.7848 ± 0.04 | 0.9924 ± 0.05 | 0.9782 ± 0.05 | 0.8765 ± 0.04 |
| MSCRED | 0.2513 ± 0.01 | 0.7319 ± 0.04 | 0.8412 ± 0.04 | 0.3741 ± 0.02 | **0.9992 ± 0.05** | 0.6770 ± 0.03 | 0.8433 ± 0.04 | 0.8072 ± 0.04 | 0.8912 ± 0.04 | 0.9862 ± 0.05 | 0.9807 ± 0.05 | 0.9363 ± 0.05 |
| MAD-GAN | 0.2233 ± 0.01 | 0.9124 ± 0.05 | 0.8026 ± 0.04 | 0.3588 ± 0.02 | 0.9593 ± 0.05 | 0.6957 ± 0.03 | 0.8463 ± 0.04 | 0.8065 ± 0.04 | 0.8516 ± 0.04 | 0.9930 ± 0.05 | 0.9862 ± 0.05 | 0.9169 ± 0.05 |
| USAD | 0.1873 ± 0.01 | 0.8296 ± 0.04 | 0.8723 ± 0.04 | 0.3056 ± 0.02 | 0.9977 ± 0.05 | 0.6879 ± 0.03 | 0.8460 ± 0.04 | 0.8143 ± 0.04 | 0.7949 ± 0.04 | 0.9912 ± 0.05 | 0.9795 ± 0.05 | 0.8822 ± 0.04 |
| MTAD-GAT | 0.2818 ± 0.01 | 0.8012 ± 0.04 | 0.8821 ± 0.04 | 0.4169 ± 0.02 | 0.9718 ± 0.05 | 0.6957 ± 0.03 | 0.8464 ± 0.04 | 0.8109 ± 0.04 | 0.7917 ± 0.04 | 0.9824 ± 0.05 | 0.9899 ± 0.05 | 0.8768 ± 0.04 |
| CAE-M | 0.2782 ± 0.01 | 0.7918 ± 0.04 | 0.8728 ± 0.04 | 0.4117 ± 0.02 | 0.9697 ± 0.05 | 0.6957 ± 0.03 | 0.8464 ± 0.04 | 0.8101 ± 0.04 | 0.7751 ± 0.04 | 1.0000 ± 0.05 | 0.9903 ± 0.05 | 0.8733 ± 0.04 |
| GDN | 0.2912 ± 0.01 | 0.7931 ± 0.04 | 0.8777 ± 0.04 | 0.4260 ± 0.02 | 0.9697 ± 0.05 | 0.6957 ± 0.03 | 0.8464 ± 0.04 | 0.8101 ± 0.04 | 0.9308 ± 0.05 | 0.9892 ± 0.05 | 0.9814 ± 0.05 | 0.9591 ± 0.05 |
| TranAD | 0.3529 ± 0.02 | 0.8296 ± 0.04 | 0.8968 ± 0.04 | 0.4951 ± 0.02 | 0.9760 ± 0.05 | 0.6997 ± 0.04 | 0.8491 ± 0.04 | 0.8151 ± 0.04 | 0.9038 ± 0.05 | **0.9999 ± 0.05** | 0.9916 ± 0.05 | 0.9494 ± 0.05 |
| Anomaly-Tran | NA | NA | NA | NA | 0.9155 ± 0.05 | 0.9673 ± 0.05 | NA | 0.9407 ± 0.05 | 0.9209 ± 0.05 | 0.9515 ± 0.05 | NA | 0.9359 ± 0.05 |
| Anomaly-Bert | NA | NA | NA | 0.5800 ± 0.02 | NA | NA | NA | 0.8540 ± 0.04 | NA | NA | NA | 0.3020 ± 0.02 |
| Plain Transformer | **0.8100 ± 0.02** | **0.6800 ± 0.02** | **0.8970 ± 0.02** | **0.7390 ± 0.02** | 0.9550 ± 0.05 | 0.9750 ± 0.04 | 0.9950 ± 0.04 | 0.9650 ± 0.04 | 0.3610 ± 0.02 | 0.7000 ± 0.03 | 0.7600 ± 0.03 | 0.4760 ± 0.02 |
| CARLA | 0.1850 ± 0.01 | 0.7316 ± 0.04 | NA | 0.2953 ± 0.01 | 0.9886 ± 0.05 | 0.5673 ± 0.03 | NA | 0.7209 ± 0.04 | 0.3891 ± 0.02 | 0.7959 ± 0.04 | NA | 0.5227 ± 0.03 |
| Our Model 1 (Copula) | 0.2887 ± 0.01 | 0.8227 ± 0.04 | 0.7580 ± 0.04 | 0.4244 ± 0.02 | 1.0242 ± 0.05 | 0.9903 ± 0.05 | 0.9950 ± 0.05 | 0.9938 ± 0.05 | **0.9794 ± 0.05** | 0.9903 ± 0.05 | 0.9936 ± 0.05 | 0.9923 ± 0.05 |
| Our Model 2 (Multivariate) | 0.2776 ± 0.01 | 0.7912 ± 0.04 | 0.7254 ± 0.04 | 0.4110 ± 0.02 | 0.9907 ± 0.05 | **0.9969 ± 0.05** | **0.9999 ± 0.05** | **0.9979 ± 0.05** | 0.9397 ± 0.05 | 0.9956 ± 0.05 | **0.9999 ± 0.05** | **0.9668 ± 0.05** |

| Method | SMD | | | | SMAP | | | |
|---|---|---|---|---|---|---|---|---|
| | P | R | AUC | F1 | P | R | AUC | F1 |
| MERLIN | 0.2871 ± 0.01 | 0.5804 ± 0.03 | 0.7158 ± 0.04 | 0.3842 ± 0.02 | 0.1577 ± 0.01 | 0.9999 ± 0.05 | 0.7426 ± 0.04 | 0.2725 ± 0.01 |
| LSTM-NDT | 0.9736 ± 0.05 | 0.8440 ± 0.04 | 0.9671 ± 0.04 | 0.9042 ± 0.04 | 0.8523 ± 0.04 | 0.7326 ± 0.04 | 0.8602 ± 0.04 | 0.7879 ± 0.04 |
| DAGMM | 0.9103 ± 0.05 | 0.9914 ± 0.05 | 0.9954 ± 0.05 | 0.9491 ± 0.05 | 0.8069 ± 0.04 | 0.9891 ± 0.05 | 0.9885 ± 0.05 | 0.8888 ± 0.04 |
| OmniAnomaly | 0.8881 ± 0.04 | 0.9985 ± 0.05 | 0.9946 ± 0.05 | 0.8414 ± 0.04 | 0.8130 ± 0.04 | 0.9419 ± 0.05 | 0.9889 ± 0.05 | 0.8728 ± 0.04 |
| MSCRED | 0.7276 ± 0.04 | 0.9974 ± 0.05 | 0.9921 ± 0.05 | 0.8414 ± 0.04 | 0.8175 ± 0.04 | 0.9216 ± 0.05 | 0.9821 ± 0.05 | 0.8664 ± 0.04 |
| MAD-GAN | **0.9991 ± 0.05** | 0.8440 ± 0.04 | 0.9933 ± 0.05 | 0.9495 ± 0.05 | 0.8157 ± 0.04 | 0.9216 ± 0.05 | 0.9891 ± 0.05 | 0.8915 ± 0.04 |
| USAD | 0.9060 ± 0.05 | 0.9974 ± 0.05 | 0.9933 ± 0.05 | 0.9495 ± 0.05 | 0.7480 ± 0.04 | 0.9627 ± 0.05 | 0.9890 ± 0.05 | 0.8419 ± 0.04 |
| MTAD-GAT | 0.8210 ± 0.04 | 0.9215 ± 0.05 | 0.9921 ± 0.05 | 0.8683 ± 0.04 | 0.7991 ± 0.04 | 0.9991 ± 0.05 | 0.9844 ± 0.05 | 0.8880 ± 0.04 |
| CAE-M | 0.9082 ± 0.05 | 0.9671 ± 0.05 | 0.9783 ± 0.05 | 0.9367 ± 0.05 | 0.8193 ± 0.04 | 0.9567 ± 0.05 | 0.9901 ± 0.05 | 0.8827 ± 0.04 |
| GDN | 0.7170 ± 0.04 | 0.9974 ± 0.05 | 0.9924 ± 0.05 | 0.8342 ± 0.04 | 0.8195 ± 0.04 | 0.9312 ± 0.05 | **0.9981 ± 0.05** | 0.8695 ± 0.04 |
| TranAD | 0.9262 ± 0.05 | 0.9974 ± 0.05 | **0.9974 ± 0.05** | 0.9605 ± 0.05 | 0.7480 ± 0.04 | 0.9891 ± 0.05 | 0.9864 ± 0.05 | 0.8518 ± 0.04 |
| Anomaly-Tran | 0.8940 ± 0.04 | 0.9545 ± 0.05 | NA | 0.9233 ± 0.05 | 0.9413 ± 0.05 | 0.9940 ± 0.05 | NA | 0.9669 ± 0.05 |
| Anomaly-Bert | NA | NA | NA | 0.5350 ± 0.03 | NA | NA | NA | 0.4570 ± 0.02 |
| Plain Transformer | 0.7420 ± 0.02 | 0.8370 ± 0.02 | 0.9500 ± 0.02 | 0.8000 ± 0.02 | 0.324 ± 0.05 | 0.727 ± 0.04 | 0.700 ± 0.04 | 0.448 ± 0.02 |
| CARLA | 0.4276 ± 0.02 | 0.6362 ± 0.03 | NA | 0.5114 ± 0.03 | 0.3944 ± 0.02 | 0.8040 ± 0.04 | NA | 0.5292 ± 0.03 |
| Our Model 1 (Copula) | 0.9985 ± 0.05 | 0.9956 ± 0.05 | 0.9904 ± 0.05 | **0.9902 ± 0.05** | **0.9984 ± 0.05** | **0.9902 ± 0.05** | 0.9902 ± 0.05 | **0.9989 ± 0.05** |
| Our Model 2 (Multivariate) | 0.9751 ± 0.05 | **0.9998 ± 0.05** | 0.9934 ± 0.05 | 0.9873 ± 0.05 | 0.9795 ± 0.05 | 0.9923 ± 0.05 | 0.9841 ± 0.05 | 0.9859 ± 0.05 |

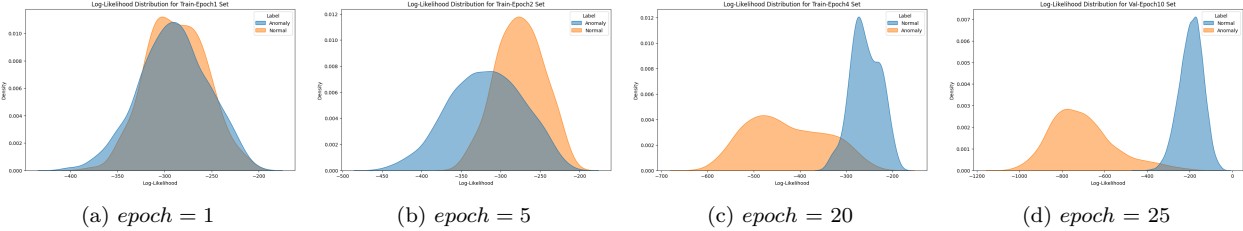

Figure 2: We jointly train the transformer and copula (student-t) parameters in the latent space with the contrastive loss and plot the normal and anomaly likelihoods. We see the gradual separation in likelihoods as we train for more epochs using our method.

## 3.2  Training Strategy

We train our network by jointly optimizing the Transformer encoder parameters, denoted $\theta$, and the joint density parameters, denoted $\phi$, in a single end-to-end fashion. First, each multivariate time series snippet is passed through the Transformer encoder, which produces a latent embedding or feature space $\mathbf{z}(\theta)$ capturing relevant temporal and cross-feature dependencies. We then evaluate the copula log-density $\log c_\phi\big(\mathbf{z}(\theta)\big)$ to quantify how well the latent feature space aligns with the joint dependency structure learned from normal data. During training, we formulate a contrastive loss that *maximizes* $\log c_\phi\big(\mathbf{z}(\theta)\big)$ for normal frames while *pushing* anomalous frames below a specified margin in log-density space. Specifically, for each normal snippet we minimize $-\log c_\phi\big(\mathbf{z}(\theta)\big)$, thus maximizing the copula likelihood, and for each anomalous snippet we include a term that penalizes its log-density if it is not sufficiently lower than the normal average. By backpropagating through both the Transformer and the copula parameters, the system iteratively updates $\theta$ and $\phi$ such that normal data cluster in a high-likelihood region, while anomalies are assigned lower density and thus become well separated in the latent feature space. Figure 1 presents a schematic illustration of our model. Algo 3 details the training procedure.

## 3.3  Ablation Study

In our ablation study, we systematically vary several hyperparameters to observe their impact on anomaly detection performance. First, we adjust the *margin* $\delta$ in our contrastive loss, finding that too small a margin may allow anomaly frames to encroach on normal regions, while too large a margin can over-penalize borderline anomalies and lead to increased false positives. We also vary the *percentage of anomaly data* we inject or label during training, confirming that a higher anomaly fraction generally helps the model to better distinguish anomalies, although an unrealistically high percentage may degrade generalization to real, rarer anomalies.

Furthermore, we experiment with different multivariate likelihood and copula families in the latent feature space—for instance, a *Gaussian* copula, a *Student-t* copula (better for heavy-tailed dependencies). We also vary *window size* and *overlap* in the construction of time-series frames, observing that larger windows capture extended patterns but raise computational cost, while more overlap provides smoother coverage but increases data redundancy. Finally, we alter the *Transformer encoder architecture* by changing the number of layers and attention heads. Other results are provided in Table 1. In Figure 2d, we observe that as training progresses, the model achieves increasingly better separation between the likelihoods of the normal and anomalous instances in the latent space. More details on the hyperparameters can be found in the appendix section.

## 3.4  Anomaly Detection Threshold Selection

In this approach, we first derive a *log-likelihood score* for each time series frame by evaluating $\log c_\phi\big(\mathbf{z}(\theta)\big)$, where $c_\phi$ is the joint density function and $\mathbf{z}(\theta)$ is the latent embedding returned by our Transformer encoder. Intuitively, a frame with higher (less negative) log-likelihood aligns better with normal behavior, while lower log-likelihood indicates a potential anomaly. We gather all log-likelihoods from the validation set and

systematically scan a range of possible thresholds, from the minimum to the maximum observed score. For each candidate threshold $\tau$, we label a frame as anomalous if its log-likelihood is below $\tau$, and we compute the corresponding F1 score on the validation data. We then select the threshold that *maximizes* the F1, thus balancing precision and recall most effectively. Once this threshold is determined, any future frame whose log-likelihood drops below $\tau$ is deemed anomalous, while frames exceeding $\tau$ are considered normal. [1]

# 4 Gradient Computation and Backpropagation

We focus on computing gradients of the contrastive loss with respect to $\phi$ (the spatial parameters that capture joint dependency) and $\theta$ (the Transformer encoder parameters that generate latent embeddings). Here, each *frame*—a short subsequence of the time-series—is treated as an individual item, potentially labeled 0 (normal) or 1 (anomalous).

---

**Algorithm 3** Single–batch update of Transformer encoder $\theta$ and latent dependency parameters $\phi$

---

**Input** : Mini-batch $\mathcal{B} = \{(x^{(i)}, y^{(i)})\}_{i=1}^N$ with $y^{(i)} \in \{0,1\}$
**Output:** Updated encoder and dependency parameters $\theta, \phi$ respectively.

1. Initialise learning rate $\eta$, margin weight $\gamma$, and standardisation parameters $(\mu, \sigma)$.

2. **Forward pass:** for each $i = 1, \ldots, N$:

   - Encode: $\mathbf{z}^{(i)} \leftarrow f_\theta^{\text{enc}}(x^{(i)})$

   - Standardise: $\mathbf{z}_{\text{std}}^{(i)} \leftarrow (\mathbf{z}^{(i)} - \mu)/\sigma$

   - Apply probability–integral transform to convert to uniform marginals: $u_k^{(i)} \leftarrow G_k\big(z_{\text{std},k}^{(i)}\big), \ k = 1, \ldots, d$

   - Compute log-likelihood density: $\ell^{(i)} \leftarrow -\log c\big(\mathbf{u}^{(i)}, \phi\big)$

   - Compute contrastive loss with soft margin:

   $$\ell^{(i)} \leftarrow \ell^{(i)} + \gamma \max\big\{0, \ \log c(\mathbf{u}^{(i)}, \phi) - (\mu_{\text{norm}} - \delta)\big\}.$$

3. Average total loss: $\mathcal{L}(\theta, \phi) \leftarrow \dfrac{1}{N} \sum\limits_{i=1}^N \ell^{(i)}$.

4. Backpropagation:

   - Latent space dependency gradient: $\nabla_\phi \mathcal{L} = -\dfrac{1}{N} \sum\limits_{i=1}^N \dfrac{\partial}{\partial \phi} \log c\big(\mathbf{u}^{(i)}, \phi\big)$

   - Encoder gradient: $\nabla_\theta \mathcal{L} = \sum\limits_{i=1}^N \dfrac{\partial \mathcal{L}}{\partial \mathbf{z}^{(i)}} \dfrac{\partial \mathbf{z}^{(i)}}{\partial \theta}$

5. Parameter updates: $\theta \leftarrow \theta - \eta \nabla_\theta \mathcal{L}, \quad \phi \leftarrow \phi - \eta \nabla_\phi \mathcal{L}$.
   **return** $\theta, \phi$

---

By treating entire frames rather than individual timesteps, we more naturally capture local temporal context, while the joint optimization of $(\theta, \phi)$ encourages both a suitable embedding space and an accurate copula-based dependency model. Anomalies emerge as frames that fail to fit this latent dependency pattern, receiving lower log-likelihood under $c(\mathbf{u}, \phi)$ and thus incurring higher contrastive penalty.

---

[1] https://github.com/padmaksha18/DACLM

## 5 Hyperparameter Tuning and Results

In the below section, we experiment with the different values of the hyperparameters and validate our model performance. In Figure 3 our experiments reveal that the classification performance of our model is significantly influenced by the selected copula family used to capture the joint dependency of the latent variables. This finding suggests that the Student-t copula provides a more accurate representation of the data-generating process, characterized by heavy-tailed distributions, rather than the Gaussian distribution. We keep the other hyper-parameters like margin, batch-size, window-length fixed for all the cases.

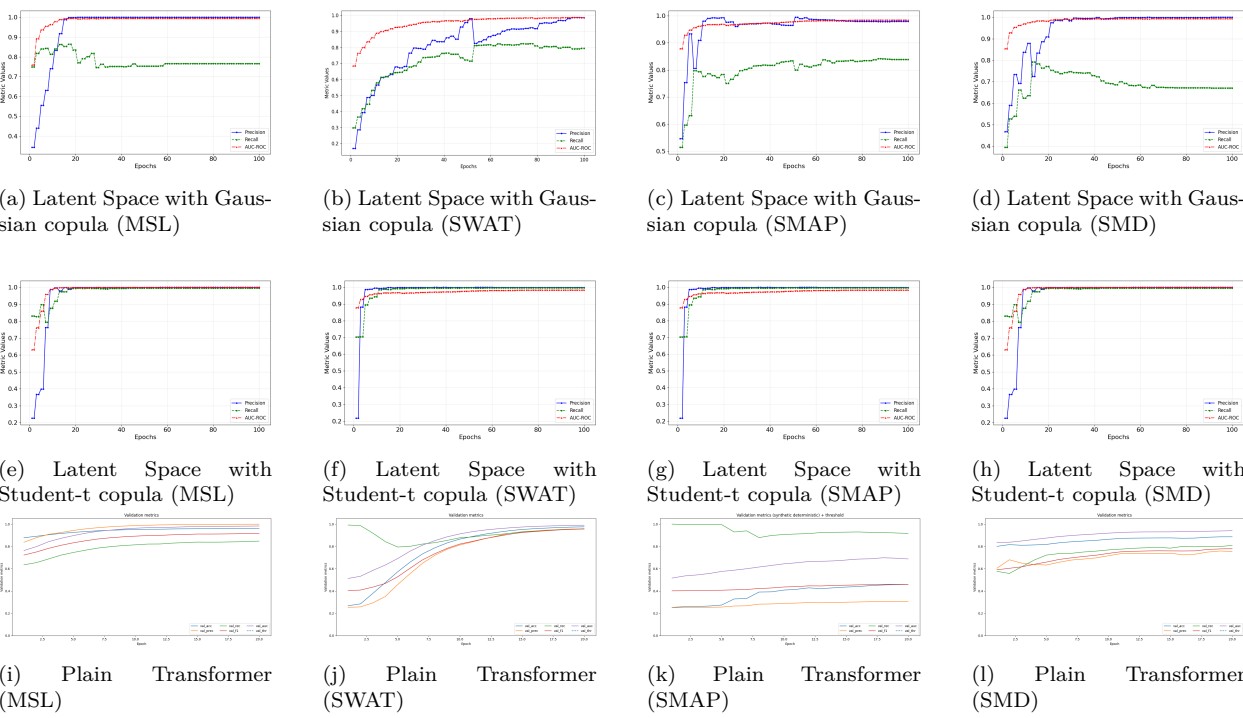

(a) Latent Space with Gaussian copula (MSL)

(b) Latent Space with Gaussian copula (SWAT)

(c) Latent Space with Gaussian copula (SMAP)

(d) Latent Space with Gaussian copula (SMD)

(e) Latent Space with Student-t copula (MSL)

(f) Latent Space with Student-t copula (SWAT)

(g) Latent Space with Student-t copula (SMAP)

(h) Latent Space with Student-t copula (SMD)

(i) Plain Transformer (MSL)

(j) Plain Transformer (SWAT)

(k) Plain Transformer (SMAP)

(l) Plain Transformer (SMD)

Figure 3: Performance metrics (Precision, recall, AUC-ROC) over epochs on the validation data when the latent space is modeled with the Gaussian Copula, Student-t Copula, and Plain Transformer for different datasets. We select the threshold based on the method described in section 3.4

Our model achieves, on average, a 5–18% improvement in classification metrics over baseline methods on most datasets. With the WADI dataset, the performance degradation is likely due to an exception to our assumption that variables exhibit distinct joint dependency structures under normal and anomalous conditions(refer to the appendix section for the detail analysis). We also observe poor performance of other baselines on the same dataset. Through ablation in Figure 3, we show that incorporating a Student-t copula—capable of modeling heavy-tailed and non-linear dependencies—leads to significant gains. Distributional analysis confirms that the feature marginals are highly skewed and deviate from Gaussianity, necessitating more expressive copula models.

## 6 Conclusion:

In our paper, we have created a unified, end-to-end anomaly detection framework that captures both temporal and multivariate relationships in complex time series. The joint modeling helps to capture the complex time-varying spatio-temporal non-linear correlations which are useful indicators of multi-variate anomalies. This approach yields a latent representation guided by copula likelihoods, effectively separating normal frames (high likelihood) from anomalous ones (low likelihood). Future work may explore more efficient attention mechanisms, advanced mixture copulas for even richer tail behaviors to improve anomaly detection in the latent space.

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

# A    Appendix

**Copula Theory**

**1. Copula Function:** Copulas allow us to represent the joint distribution of random variables using their marginal distributions and a copula function that captures their dependency structure. The joint PDF can be written as:

$$p(\mathbf{z}) = c(\mathbf{u}; \phi) \prod_{i=1}^{d} p(z_i),$$

where $\mathbf{u} = (u_1, u_2, \ldots, u_d)$ with $u_i = F_{Z_i}(z_i)$ (the marginal CDF of $Z_i$), and $c(\mathbf{u}; \phi)$ is the copula density.

**2. Dependency Structure:** The copula density $c(\mathbf{u}; \phi)$ encapsulates the dependency structure among the variables, while the marginal distributions $p(z_i)$ account for their individual behaviors.

### Rewriting Mutual Information Using Copulas

Substituting the copula representation of $p(\mathbf{z})$ into the mutual information formula:

$$I(\mathbf{Z}) = \int p(\mathbf{z}) \log \left( \frac{c(\mathbf{u}; \phi) \prod_{i=1}^{d} p(z_i)}{\prod_{i=1}^{d} p(z_i)} \right) d\mathbf{z}.$$

Simplifying the logarithmic term:

$$I(\mathbf{Z}) = \int p(\mathbf{z}) \log c(\mathbf{u}; \phi) \, d\mathbf{z}.$$

### Change of Variables From z to u

To simplify the integral, we perform a change of variables:

$$u_i = F_{Z_i}(z_i), \quad z_i = F_{Z_i}^{-1}(u_i).$$

The Jacobian determinant of this transformation is given by:

$$|J| = \prod_{i=1}^{d} p(z_i).$$

Thus, the volume element transforms as:

$$d\mathbf{z} = \frac{d\mathbf{u}}{|J|} = \frac{d\mathbf{u}}{\prod_{i=1}^{d} p(z_i)}.$$

Substituting this change of variables into the integral:

$$I(\mathbf{Z}) = \int c(\mathbf{u}; \phi) \log c(\mathbf{u}; \phi) \, d\mathbf{u}.$$

### Final Expression for Mutual Information

The mutual information among the variables $\mathbf{Z}$ is expressed in terms of the copula density as:

$$I(\mathbf{Z}) = \int c(\mathbf{u}; \phi) \log c(\mathbf{u}; \phi) \, d\mathbf{u}.$$

This formulation shows that mutual information is equivalent to the expected log-likelihood of the copula density. The copula density $c(\mathbf{u}; \phi)$ captures the dependency structure among the variables, independent of their marginal distributions, providing a comprehensive measure of dependency.

## B   Why Latent Space -computation complexity and high-dimensional setting

We implement Gaussian and Student-$t$ copulas in a low-dimensional latent space by parameterizing the correlation matrix $\Sigma = LL^T$ using its Cholesky factor $L$, with positive diagonals enforced via a `softplus` function. Latent variables are standardized, mapped through the Student-$t$ CDF, and transformed back using the inverse CDF (PPF). The log-determinant is computed as $\log |\Sigma| = 2 \sum \log(L_{ii})$ in $O(d)$ time, and the quadratic form $x^T \Sigma^{-1} x$ is evaluated via einsum operations with $O(d^2)$ complexity. Although Cholesky inversion has a worst-case cost of $O(d^3)$, the small latent dimension $d$ makes these operations computationally efficient. In our case, with datasets having fewer than 100 features, the added overhead is minimal. For high-dimensional settings, Salinas et al. (2019) propose decomposing the covariance matrix into a diagonal and a low-rank component, reducing parameters from $O(N^2)$ to $O(Nr)$, and overall complexity to $O(Nr^2)$, which is effectively $O(N)$ for small fixed $r$.

## B.1 Latent Space Analysis

The latent space analysis reveals that projecting data into a narrow bottleneck (i.e., low-dimensional space) leads to degraded model performance. In contrast, the model performs significantly better in higher-dimensional latent spaces, such as 64 or 128 dimensions (Figure 4).

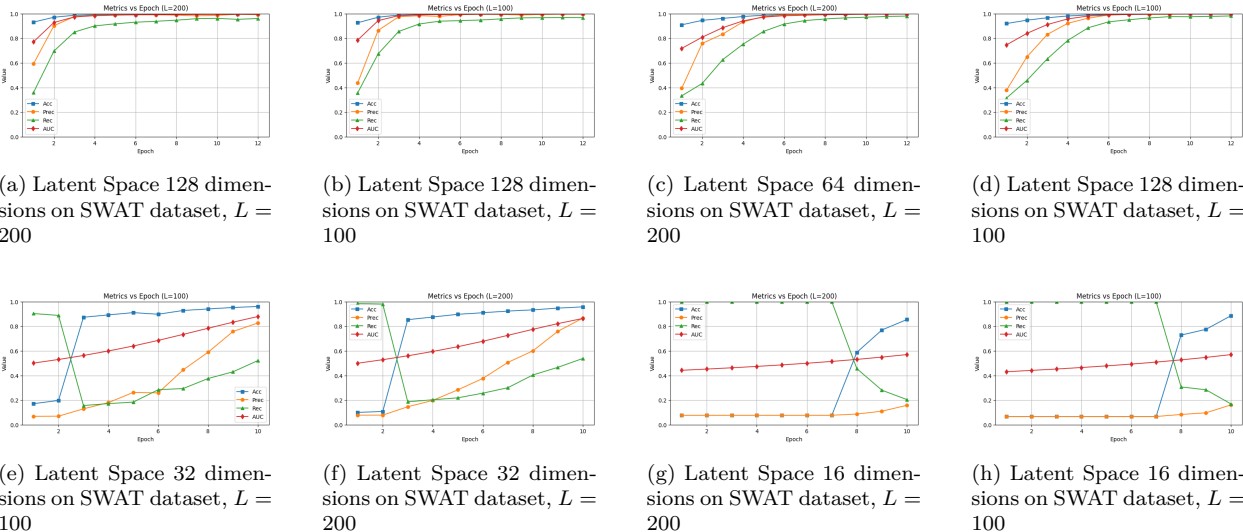

(a) Latent Space 128 dimensions on SWAT dataset, $L = 200$

(b) Latent Space 128 dimensions on SWAT dataset, $L = 100$

(c) Latent Space 64 dimensions on SWAT dataset, $L = 200$

(d) Latent Space 128 dimensions on SWAT dataset, $L = 100$

(e) Latent Space 32 dimensions on SWAT dataset, $L = 100$

(f) Latent Space 32 dimensions on SWAT dataset, $L = 200$

(g) Latent Space 16 dimensions on SWAT dataset, $L = 200$

(h) Latent Space 16 dimensions on SWAT dataset, $L = 100$

Figure 4: Performance metrics (Precision, recall, AUC-ROC) over epochs on the validation data when the latent space dimension is changed. The top row is for higher dimensions 128 and 64, the bottom row is for lower dimensions 32 and 16 for different context lengths $L$

## B.2 T-SNE projection of the latent space

Here, we visualize the test or validation data by projecting it into the trained latent space using t-SNE, which reduces high-dimensional representations to two dimensions while preserving local neighborhood structure for better interpretability. However, we believe that unlike the likelihood score, t-SNE preserves neighborhood ranks, not absolute density, and may place normal and anomalous points close together despite vastly different likelihoods. Additionally, in high dimensions, pairwise distances become unreliable due to the concentration phenomenon, making t-SNE embeddings unstable. In contrast, the copula likelihood leverages full joint density, providing a more reliable measure for anomaly detection (Figure 5).

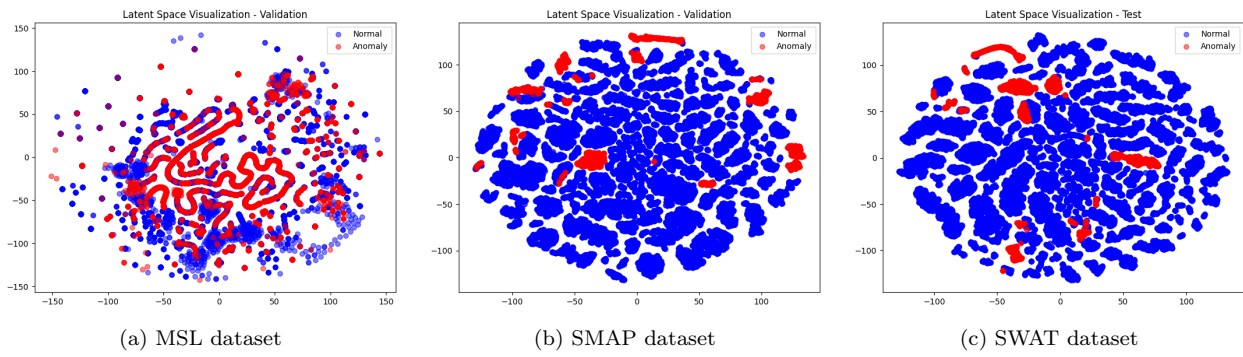

(a) MSL dataset

(b) SMAP dataset

(c) SWAT dataset

Figure 5: Latent space projections using t-SNE on various datasets.

## C  Hyperparameters

The following are the key hyperparameters on which our model performance depends.

*Window Size (L):* Each multivariate time-series snippet is of length $L$, meaning we process $L$ consecutive timestamps as a single input frame. This choice controls how much local context the model sees at once.

*Overlap or Step Length:* After extracting a window of length $L$, we often shift by a smaller step (e.g., $L/2$ or another fraction) for the next window, creating overlapping frames. Overlaps help capture transitions more smoothly and increase data availability, but can lead to redundancy if the overlap is too large.

*Number (or Percentage) of Anomalies:* We typically define what fraction (e.g., 10% or 30%) of frames to label as anomalous for synthetically generated sets. We gradually vary this percentage and check the model performance with various percentages of synthetic anomaly samples.

*Batch Size (B):* The batch size indicates how many frames we process in one forward/backward pass.

*Transformer Depth and Heads:* Although not strictly a hyperparameter, we experiment with different numbers of layers, attention heads, and embedding dimensions.

*Copula Family and Parameters:* In the latent feature space, we choose a specific copula family (Gaussian, Student-t) along with its parameters or estimation method. This affects how well the model captures the joint dependency structure among latent feature dimensions.

*Margin (δ) in Contrastive Loss:* We define a margin $\delta$ that ensures that anomalies remain sufficiently below the normal log-likelihood region. Larger $\delta$ forces stricter separation but can cause more false alarms if the model over-penalizes borderline samples. We employ a sample-level margin, ensuring that each anomaly sample lies below that margin.

*Learning Rate and Optimizer:* Finally, we choose an optimizer (e.g., Adam) and a learning rate for both the Transformer encoder parameters and the copula parameters. Tuning this is crucial to ensure stable, efficient convergence.

*Weightage (α):* We define the $\alpha$ parameter to balance the different components of our loss function. We experiment with different values of $\alpha$.

## D  Average Detection Delay (ADD) Computation and Interpretation

Let $\{y_t\}_{t=1}^{T}$ be the ground-truth labels (0=normal, 1=anomalous). We define each anomaly *event* as a contiguous run of ones; its true start $t_s$ is the first index where $y_{t_s} = 1$ and $y_{t_s-1} = 0$. Using sliding windows of length $L$ (stride=1), window $i$ covers samples $[i, \ldots, i + L - 1]$ and thus *ends* at

$$t_{\text{end},i} = i + L - 1.$$

After thresholding log-likelihoods we obtain binary predictions $p_i \in \{0, 1\}$. For each event start $t_s$, we collect all flagged windows with $p_i = 1$ whose end $t_{\text{end},i} \geq t_s$, and set the detection time

$$t_d = \min\{ t_{\text{end},i} \mid p_i = 1, \ t_{\text{end},i} \geq t_s \}.$$

The per-event detection delay is then

$$\Delta = t_d - t_s.$$

The Average Detection Delay (ADD) is the mean of these per-event delays over $N$ events:

$$\text{ADD} = \frac{1}{N} \sum_{k=1}^{N} \Delta_k.$$

ADD measures the average number of time-steps between the true onset of an anomaly and the model's first alarm. A lower ADD indicates quicker detection. For example, ADD = 20 means that, on average, the detector flags an event 20 time-steps after its actual start.

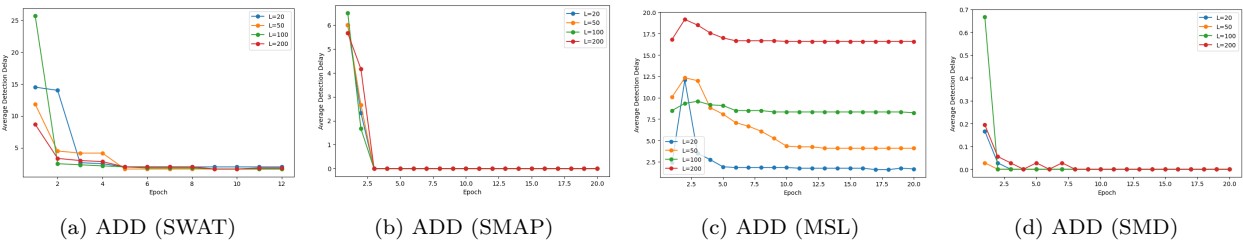

| (a) ADD (SWAT) | (b) ADD (SMAP) | (c) ADD (MSL) | (d) ADD (SMD) |

Figure 6: We plot the ADD vs epochs on various datasets considering different context lengths($L$) on the validation data during training

In the ADD-vs-epoch plot, we observe that ADD decreases steadily during training and converges to zero for all window lengths $L$ for most datasets. By mid-training, the model consistently flags the anomaly in the very first eligible window (ending at $t_s$), yielding zero delay. This behavior confirms that the model learns to react immediately at the first sign of an anomaly as training progresses (Figure 6).

## E  Marginal and Joint Dependency Perturbation Procedure on Synthetic Data

To synthesize anomalies that exhibit both marginal shifts and joint-dependency changes, we apply the following sequence of perturbations to the latent features $Z \in \mathbb{R}^{T \times d}$:

1. **Affine Warp:**
$$Z'_{t,j} = s_j\, Z_{t,j} + b_j, \quad s_j \sim \mathcal{U}(s_{\min}, s_{\max}),\ b_j \sim \mathcal{U}(b_{\min}, b_{\max})$$

   where $(s_{\min}, s_{\max})$ and $(b_{\min}, b_{\max})$ control the strength of scale-and-shift.

2. **Sequential Nonlinear Warps:** Apply $K$ random curvature transforms in series:

$$Z'' = f_{i_K}\bigl(\ldots f_{i_2}\bigl(f_{i_1}(Z')\bigr)\ldots\bigr),$$

   where each $f_i$ is one of

   - Power warp: $x \mapsto \operatorname{sign}(x)\,|x|^p,\ p \sim \mathcal{U}(p_{\min}, p_{\max})$.
   - Log-soft warp: $x \mapsto \operatorname{sign}(x)\,\ln\bigl(1 + a\,|x|\bigr),\ a \sim \mathcal{U}(a_{\min}, a_{\max})$.
   - Sinh warp: $x \mapsto \sinh(b\,x),\ b \sim \mathcal{U}(b_{\min}, b_{\max})$.

3. **Heteroskedastic Noise:** Add time-varying Gaussian noise with amplitude modulated by a sinusoid:
$$Z'''_{t,j} = Z''_{t,j} + \varepsilon_{t,j}, \quad \varepsilon_{t,j} \sim \mathcal{N}\bigl(0, \sigma_t^2\bigr), \quad \sigma_t = \sigma\bigl(1 + A\sin(2\pi t/T)\bigr),$$
   ensuring $\sigma_t \geq 0$ by clipping.

By introducing affine and nonlinear warps plus heteroskedastic noise and heavy-tails, we force a multivariate Gaussian or Student-$t$ density— which depends on correct marginal forms— to react strongly to pure marginal distortions. In contrast, a copula-based detector relies only on rank-correlations and is *invariant* to these marginal changes, so it remains focused on true joint-dependency shifts from $B_{\mathrm{norm}}$ to $B_{\mathrm{anom}}$, yielding robustness under minor to severe marginal perturbations.

### E.1  Parameter Definitions and Evaluation Cases

`dependency_strength` $\alpha$ Base coupling strength of the latent transition matrix $B_{\mathrm{norm}}$; controls how strongly features co-evolve in the normal regime.

`anomaly_dependency_shift` $\delta$ Magnitude of change from $B_{\mathrm{norm}}$ to $B_{\mathrm{anom}}$; larger $\delta$ induces a bigger joint-dependency shift for anomalies.

*Marginal change* Degree of distortion applied to marginals via affine/nonlinear warps, and heteroskedastic noise.

1. **Case 1: Balanced (moderate) marginals and joint dependencies**

   - moderate marginal shift.
   - moderate joint shift.
   - Apply moderate warp parameters and noise amplitudes to distort marginals.

| Window Length ($L$) | Plain Transformer | | | | Student-t Multivariate Model | | | | Student-t Copula Model | | | |
|---|---|---|---|---|---|---|---|---|---|---|---|---|
| | Accuracy | Precision | Recall | AUC | Accuracy | Precision | Recall | AUC | Accuracy | Precision | Recall | AUC |
| $L = 20$ | 0.514 | 0.324 | 0.866 | 0.727 | 0.934 | 0.950 | 0.917 | 0.983 | 0.981 | 0.981 | 0.981 | 0.998 |
| $L = 50$ | 0.517 | 0.324 | 0.856 | 0.725 | 0.919 | 0.933 | 0.904 | 0.974 | 0.978 | 0.980 | 0.975 | 0.995 |
| $L = 100$ | 0.538 | 0.334 | 0.849 | 0.735 | 0.920 | 0.941 | 0.895 | 0.972 | 0.974 | 0.988 | 0.959 | 0.993 |
| $L = 200$ | 0.538 | 0.334 | 0.849 | 0.735 | 0.921 | 0.950 | 0.889 | 0.970 | 0.974 | 0.986 | 0.962 | 0.993 |

Table 2: Comparison of performance metrics between Plain Transformer, Student-t Multivariate Model, and Student-t Copula Model across different window (context) lengths $L$.

2. **Case 2: High marginal drift, strong joint dependency shift**

   - High marginal change.
   - large (strong joint-dependency shift).
   - Use extreme warp ranges and high heteroskedastic noise to maximize marginal change.

| Window Length ($L$) | Plain Transformer | | | | Student-t Multivariate Model | | | | Student-t Copula Model | | | |
|---|---|---|---|---|---|---|---|---|---|---|---|---|
| | Accuracy | Precision | Recall | AUC | Accuracy | Precision | Recall | AUC | Accuracy | Precision | Recall | AUC |
| $L = 20$ | 0.416 | 0.284 | 0.876 | 0.629 | 0.917 | 0.911 | 0.924 | 0.976 | 0.990 | 0.992 | 0.989 | 0.999 |
| $L = 50$ | 0.458 | 0.294 | 0.836 | 0.633 | 0.892 | 0.899 | 0.883 | 0.959 | 0.988 | 0.995 | 0.980 | 0.998 |
| $L = 100$ | 0.446 | 0.290 | 0.839 | 0.622 | 0.880 | 0.896 | 0.859 | 0.952 | 0.986 | 0.997 | 0.976 | 0.998 |
| $L = 200$ | 0.504 | 0.309 | 0.796 | 0.610 | 0.884 | 0.917 | 0.846 | 0.952 | 0.988 | 0.994 | 0.982 | 0.998 |

Table 3: Comparison of performance metrics between Plain Transformer, Student-t Multivariate Model, and Student-t Copula Model across different window (context) lengths $L$ (Case 2).

3. **Case 3: High marginal drift, weak joint dependency shift**

   - Marginal high change.
   - weak joint-dependency shift.
   - Use near-identity warp (e.g. small scales, zero noise) so marginals remain essentially unchanged.

| Window Length ($L$) | Plain Transformer | | | | Student-t Multivariate Model | | | | Student-t Copula | | | |
|---|---|---|---|---|---|---|---|---|---|---|---|---|
| | Accuracy | Precision | Recall | AUC | Accuracy | Precision | Recall | AUC | Accuracy | Precision | Recall | AUC |
| $L = 20$ | 0.881 | 0.749 | 0.789 | 0.923 | 0.957 | 0.962 | 0.951 | 0.991 | 0.970 | 0.971 | 0.969 | 0.995 |
| $L = 50$ | 0.874 | 0.731 | 0.783 | 0.917 | 0.908 | 0.926 | 0.887 | 0.968 | 0.956 | 0.962 | 0.951 | 0.988 |
| $L = 100$ | 0.880 | 0.784 | 0.716 | 0.913 | 0.884 | 0.881 | 0.889 | 0.952 | 0.947 | 0.968 | 0.924 | 0.976 |
| $L = 200$ | 0.875 | 0.800 | 0.800 | 0.920 | 0.879 | 0.882 | 0.876 | 0.948 | 0.936 | 0.958 | 0.912 | 0.971 |

Table 4: Comparison of performance metrics between Plain Transformer, Student-t Multivariate Model, and Student-t Copula across different window (context) lengths $L$.

Across all cases (Table 2, Table 3 and Table 4), the Student-t copula consistently outperforms the standard Student-t multivariate likelihood model, particularly when marginal distribution drifts are combined with weak to strong shifts in joint dependencies. Interestingly, the plain transformer performs better when the joint-dependency shift is weaker which highlights the importance of modeling the joint dependency separately when its strong.

## F   Transformer Encoder Configurations

Although we explored various Transformer encoder configurations, we did not observe a direct improvement in classification performance with increased model complexity, such as adding more layers or attention heads. Interestingly, a lightweight model (e.g., Config 1, Config 2) achieved the best performance in our experiments.

Table 5: Detailed Configurations of Transformer Encoder Models

| Parameter | Config 1 | Config 2 | Config 3 | Config 4 |
|---|---|---|---|---|
| Input Dimension (input_dim) | 32 | 64 | 32 | 128 |
| Model Dimension (model_dim) | 64 | 128 | 256 | 512 |
| Number of Layers (num_layers) | 4 | 6 | 8 | 12 |
| Number of Attention Heads (num_heads) | 2 | 4 | 8 | 8 |
| Dropout Rate (dropout) | 0.1 | 0.2 | 0.3 | 0.15 |
| Pooling Mode (pooling_mode) | Mean | Sum | Mean | Mean |
| Feedforward Dimension (dim_feedforward) | 128 | 256 | 512 | 1024 |

## G   Baseline Comparison (LSTMs)

Here, we replace the Transformer with an alternative sequence model (LSTM) to compare their performance. We observe that the Transformer converges faster during training and achieves better performance on the SWAT dataset. However, across other datasets, the difference between the two models is not significant. (See Figure 7 for SWAT results.)

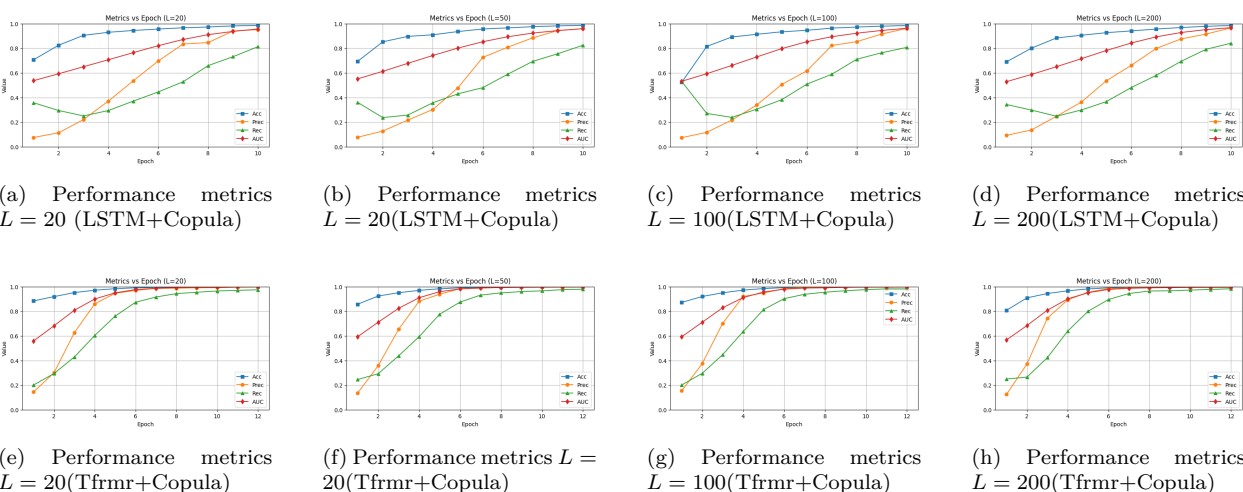

(a) Performance metrics $L = 20$ (LSTM+Copula)

(b) Performance metrics $L = 20$(LSTM+Copula)

(c) Performance metrics $L = 100$(LSTM+Copula)

(d) Performance metrics $L = 200$(LSTM+Copula)

(e) Performance metrics $L = 20$(Tfrmr+Copula)

(f) Performance metrics $L = 20$(Tfrmr+Copula)

(g) Performance metrics $L = 100$(Tfrmr+Copula)

(h) Performance metrics $L = 200$(Tfrmr+Copula)

Figure 7: Performance metrics (Precision, recall, AUC-ROC) over epochs on the validation data when the latent space is modeled with LSTM and Student-t Copula(a-d) versus Transformer and Student-t Copula on the SWAT dataset for different context lengths($L$).

## H   Dataset Details

The marginal histograms show that several SWaT features (e.g., FIT501, AIT201) exhibit clear distributional shifts between normal (blue) and anomaly (red) regimes, while others overlap more subtly. Under normal operation, the correlation heatmap reveals strong positive and negative dependencies among specific sensor pairs, forming distinct clusters. In the anomaly heatmap Figure 11, many of these dependencies weaken, invert, or disappear, indicating disrupted joint behavior. Together, these plots demonstrate that anomalies

Table 6: Details of the benchmark datasets used in our experiments.

| Dataset | Train length | Test length | Anomaly % in test | Dimension |
|---------|-------------|-------------|-------------------|-----------|
| SWaT (2017) | 495,000 | 449,919 | 12.13% | 51 |
| WADI (2017) | 784,537 | 172,801 | 5.77% | 123 |
| MSL (2018) | 58,317 | 73,729 | 10.53% | 55 |
| SMAP (2018) | 153,183 | 427,617 | 12.79% | 25 |
| SMD (2019) | 25,300 | 25,301 | 4.16% | 38 |

manifest both as marginal drifts and as changes in feature interdependence, motivating our copula-based modeling of joint distributions.

# I  Comparison of SWaT and WADI: Marginal and Dependency Characteristics

## I.1  Marginal Distributions

- **SWaT:** Most features exhibit broad, continuous histograms under both normal and anomalous regimes. Anomalies manifest as systematic shifts (new modes or heavier shoulders) rather than isolated spikes.

- **WADI:** Many sensors remain nearly constant during normal operation, producing *spiky*, low-variance marginals. Anomalies appear as single-point outliers against a Dirac-like baseline, resulting in extremely sparse, heavy-tailed histograms.

## I.2  Joint Dependency Patterns

- **SWaT:** The correlation heatmaps reveal clear block structures (clusters of mutually correlated variables) that deform smoothly under anomaly. These coherent dependency shifts are readily captured by rank-based copula estimators.

- **WADI:** The normal-state heatmap is noisy and lacks large cohesive blocks. Under anomaly, only a handful of near-constant sensors exhibit perfect correlations, while most variables remain uninformative due to zero or near-zero variance.

## I.3  Implications for Spatio-temporal Detection

- On **SWaT**, anomalies induce both marginal shifts and *consistent* joint-dependency changes across clusters. The copula component—being invariant to marginals—focuses on these dependency deformations, and the Transformer encodes their spatio-temporal patterns effectively.

- On **WADI**, the scarcity of non-constant features and the predominance of one-off marginal spikes undermine rank-correlation estimates. Copula inference becomes unstable or uninformative, and the model cannot learn a coherent dependency change to discriminate anomalies.

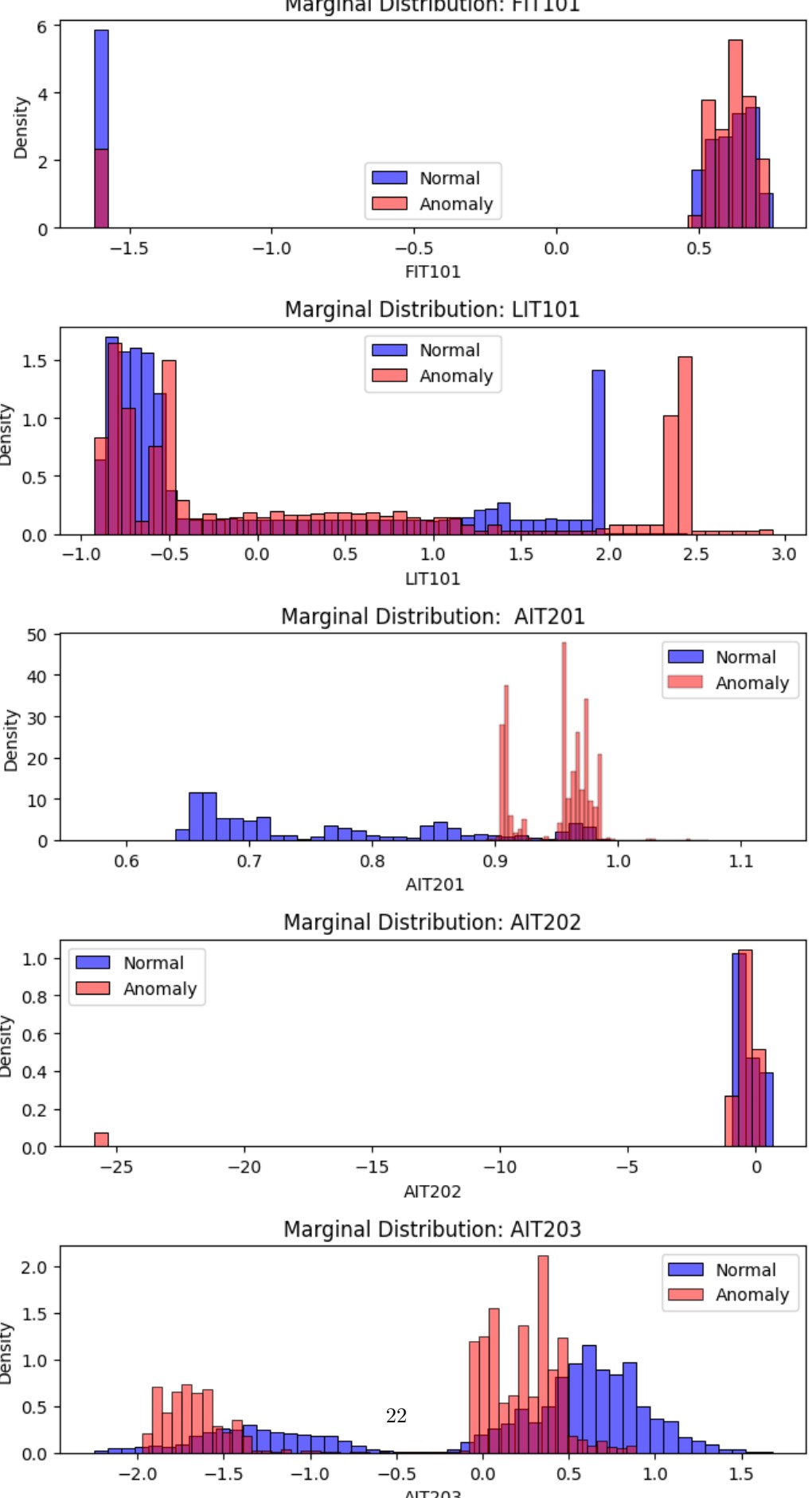

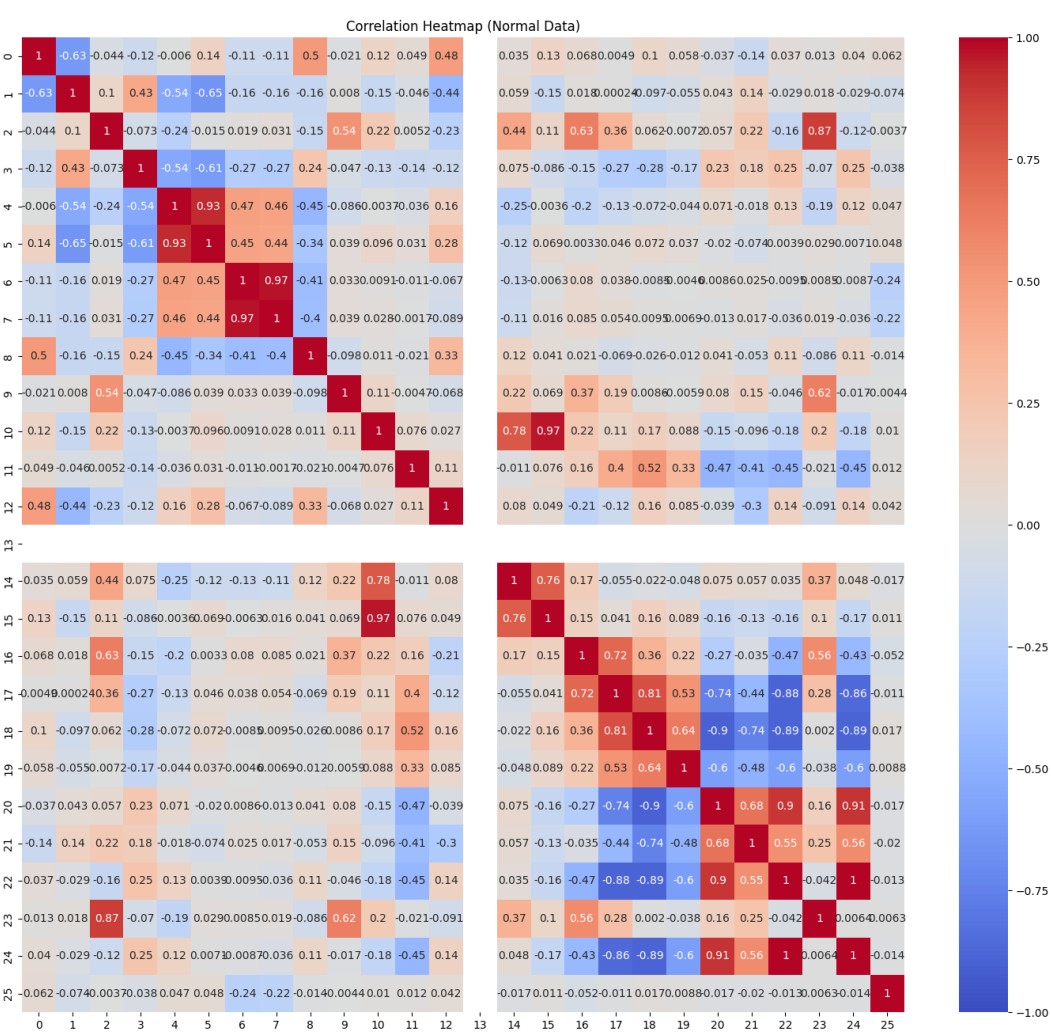

Figure 9: Correlation heatmap of SWaT features during normal operation. Distinct clusters with strong intra-cluster correlations highlight the structured dependencies among sensors.

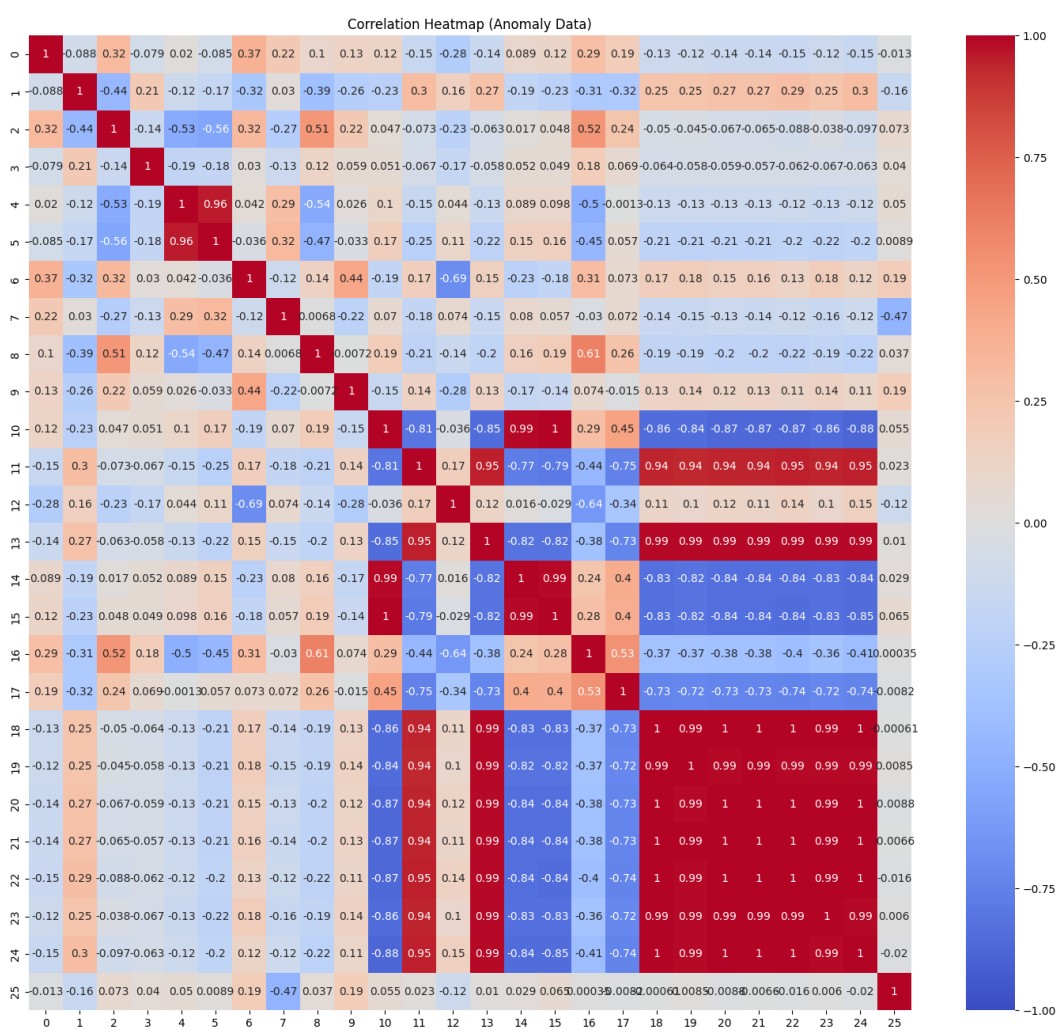

Figure 10: Correlation heatmap of SWaT features during anomalous operation. Many previously strong correlations weaken or invert, revealing disrupted joint dependencies in the anomaly regime.

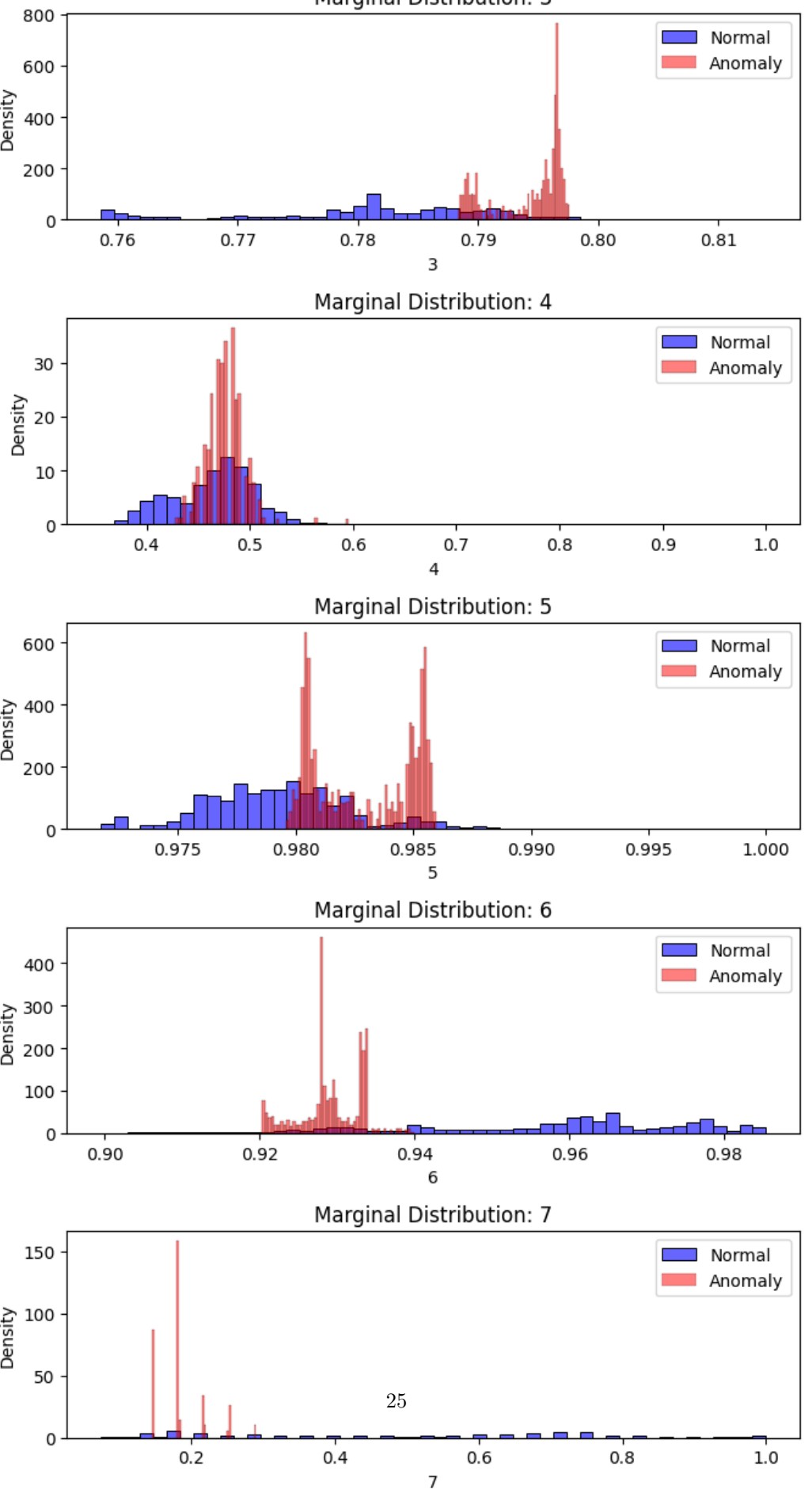

