# OpenReview forum: "Beyond Marginals: Learning Joint Spatio-Temporal Patterns for Multivariate Anomaly Detection"
_TMLR — Accepted by TMLR_

### Review · Reviewer_zeTd · 2025-06-04

**Summary Of Contributions:**

An end-to-end Transformer with a Gaussian/Student-t dependency head learns joint, not just marginal, but using a contrastive objective, benchmarked with 15 baselines on five multivariate-time-series anomaly-detection benchmarks.

Please note that the following represents my initial impressions of the paper, and I am open to discussion. I welcome any corrections to potential misunderstandings.

**Audience:**

Yes

**Broader Impact Concerns:**

I don't see critical concerns, but risk of false assurance in safety-critical monitoring could be worth discussing. The method’s current evaluation omits latency metrics and relies on a single, stationary dependence model; if deployed in time-critical application such as medical or autonomous-vehicle settings, delayed or missed detections could cause physical or financial harm. A Broader Impact Statement could discuss mitigation strategies (e.g., human-in-the-loop confirmation).

**Claims And Evidence:**

No

**Requested Changes:**

I have incorporated requests in Strengths and Weakness section, please refer to it. My most critical concern is the first three.

**Strengths And Weaknesses:**

## Strengths.
- Principled dependency module on top of a Transformer encoder, moving anomaly detection beyond purely marginal modelling.
- Convenient end-to-end framework with a simple objective, avoiding complex multi-stage pipeline.

## Weaknesses.
### Unsubstantiated motivation.
The introduction asserts that LSTMs/Transformers “often struggle to capture latent dependencies” but cites no prior evidence and supplies no diagnostic experiment isolating that failure. Without either literature support or a synthetic test case where marginals are unchanged yet joint structure shifts, the central premise remains speculative.
- Suggestion: add citations to empirical studies documenting this limitation and include a toy dataset where only cross-variable covariance changes; compare a marginal-only baseline with the proposed model to quantify the gap.

### Terminological mismatch: model is not a true copula
The encoder outputs unrestricted $\mathbf z\in\mathbb R^{d_z}$. The “copula head” then evaluates an elliptical density directly on $\mathbf z$; no probability–integral transform forces uniform marginals. Consequently the function lacks the mathematical properties of a copula.

### Sliding-window labelling may delay alarms
A window is flagged anomalous if any timestamp inside is anomalous. Near the onset of an event, the window is dominated by normal samples, potentially postponing detection by up to $L-1$ points. The paper evaluates only per-window F1/AUC, so latency cost is invisible.
- Suggestion: add average detection delay (ADD) or time-to-detect metrics and vary $L$ to show the precision–delay trade-off. Compare with a point-wise baseline to quantify latency.

### Stationarity assumption left implicit
A single, global correlation matrix is learned and then fixed at test time, implying time-stationary dependence. Section 2 never acknowledges regime shifts, concept drift, or how the model would cope with them.
- Suggestion: explicitly state the assumption; perform a drift experiment (train on regime A, test on regime B) or outline an adaptive extension (e.g. exponentially weighted covariance updates).

### Missing formal definition of a copula
The manuscript jumps directly to “Gaussian/Student-t copula” without stating the standard definition (uniform marginals on $[0,1]^d$ and Sklar’s theorem). Reviewers unfamiliar with copula theory must look elsewhere.
- Suggestion: open Section 2 with a concise formal definition with reference to Appendix A. Also place the Gaussian and Student-t specialisations there rather than in the experimental section.

### Performance claims over-state the empirical evidence
Table 1 does not show uniform dominance. While providing a state-of-the-art model is not a necessary requirement, reproducibility concern still remains.
- Suggestion: report mean ± std over multiple seeds. If possible run a statistical test to report p-value.

---

> ### Author Response · Authors · 2025-07-18
> **Clarifications addressed to Reviewer zeTd**
>
> We sincerely thank the reviewer for their thoughtful feedback, which helped us improve the clarity and depth of our work.
>
> 1. "Unsubstantiated motivation" - We have now cited relevant papers in the third paragraph of the introduction to underscore the importance of capturing inter-time series relationships—such as correlations among multiple variables—alongside temporal context for more explicit anomaly detection in multivariate time series. These works also highlight the limitations of standard deep learning models like CNNs and LSTMs in independently modeling the intertwined spatial and temporal dependencies, often necessitating more sophisticated architectures. We discussed a spatio-temporal cyber-physical system example, where a sensor reading at time $t$ may be correlated with another at time $t-1$. Salinas et al., in their work, emphasize that assuming independence is inappropriate in contexts where temporal or spatial correlations are significant. They also point out that in anomaly detection, a collective deviation across multiple nodes may signal an issue—even if no individual node displays overtly anomalous behavior.
>
>     Salinas, David, et al. "High-dimensional multivariate forecasting with low-rank gaussian copula processes." Advances in neural information processing systems 32 (2019).
>
> 2. We thank the reviewer for raising this important point. The central objective of our work is to improve anomaly detection in multivariate data by modeling the joint distribution of variables—capturing both spatial and temporal dependencies. We explore two complementary approaches: (i) modeling latent spatial correlations using multivariate Gaussian and Student-t likelihoods, and (ii) leveraging Gaussian and Student-t copulas from the copula family. These approaches are detailed separately in Algorithm 1 and Algorithm 2, with the training procedure outlined in Algorithm 3.
>
>     As a prerequisite, we transform the unknown marginals into Gaussian marginals to ensure the applicability of the Gaussian PDF—this step is essential because random vectors in their original form may lie in an arbitrary space where the Gaussian PDF is not valid. In the copula-based method (Algorithm 2), we perform a probability–integral transform to enforce uniform marginals on the latent variables $Z$ (Step 2) before computing the copula log-density, thereby satisfying the required mathematical properties of a copula.
>
> 3. "Sliding-window labelling may delay alarms" - We added section C in the Appendix and discuss the latency vs performance trade-off on all the datasets.
>
> 4. "Stationarity assumption left implicit" - We evaluate our model on real-world datasets from diverse cyber-physical systems. Since concept drift is typical in real-time streaming data, we generate synthetic data with injected concept drift to assess our model’s robustness. Detailed results are provided in the appendix section D.
>
> 5. "formal definition of a copula" - We added the mathematical notion of copulas in Section 2.3.1 followed by the algorithms we design to estimate joint dependency in the latent space. (Algo 1 and Algo 2)
>
> 6. "Performance claims over-state the empirical evidence" - We have added the statistical significance of our results in Table 1.

---

> > ### Comment · Reviewer_zeTd · 2025-07-29
> >
> > Thank you for the substantial revisions and detailed rebuttal; I appreciate the effort to make the manuscript clearer and somewhat stronger. However, I still feel that the central justification that modern Transformer-based detectors fail to model joint dependencies remains unproven. The introduction now cites three recent papers (Tian 2023, Chen 2023, Zheng 2023) but if I understand correctly, none of the new citations shows a failure of "Transformers" specifically. No diagnostic experiment was added. The new latency and drift appendices are promising yet incomplete (no false alarm trade-off, no baselines). Thus, overall, in its present form it is difficult for me to recommend acceptance.

---

> > > ### Author Response · Authors · 2025-07-30
> > >
> > > Dear Reviewer, We sincerely appreciate the opportunity to clarify the novelty of our contribution.
> > >
> > > With due respect, we would like to kindly bring to your attention that the challenge of capturing joint dependencies in modern Transformer-based detectors has indeed been previously noted, particularly in the work by Jeong et al. (AnomalyBert 23'), Tuli et al. (TranAD 22'), and Jiehui et al (Anomaly Transformer 21'). In their abstract, they highlight the importance of "understanding the temporal context and interrelation between variables simultaneously" for effective AD in multivariate time series.  AnomalyBERT introduces a specialized Transformer architecture that incorporates 1D relative position bias to better capture these multivariate spatial interactions. The authors of TranAD highlight that plain Transformer architectures struggle with detecting subtle anomalies in multivariate data, often failing when deviations are minor. To overcome this, the authors enhance Transformers using adversarial training and self-conditioning to amplify anomaly signals and improve generalization. By incorporating model-agnostic meta-learning, TranAD outperforms basic Transformer models by over 11%. In Anomaly Transformer, the authors argue that “pointwise” reconstruction or prediction errors treat each timestamp in isolation, so they fail to capture longer‐range dynamics and are easily overwhelmed by the vast majority of normal points. Because anomalies often manifest as deviations in temporal patterns rather than single‐point spikes, a windowed or contextual representation—where each score reflects an entire subsequence—can better encode trends, seasonality, and abrupt changes.
> > >
> > > In our manuscript, we acknowledge their contribution to all these works in the introduction, related work, and baselines (Table 1). We use the original Transformer model (Vaswani et al., without relative position encoding) solely as a temporal encoder. Rather than modifying the Transformer to capture spatial dependencies, our key innovation lies in integrating it with a theoretically grounded multivariate modeling framework—leveraging Student-t likelihoods and copula theory—to explicitly model nonlinear spatial dependencies.
> > >
> > > We directly compare our method with all these baselines in Table 1, where both of our model variants outperform them on four out of five datasets. Furthermore, the recently added references (Tian 2023; Chen 2023; Zheng 2023) reinforce the broader point that conventional sequence models often fall short in modeling latent variable dependencies, particularly under nonlinear or complex inter-variable relationships. While not specific to Transformer architectures, these works support our motivation.
> > >
> > > Our central contribution is a unified spatio-temporal modeling framework that combines Transformers (for temporal context) with copula-based modeling of multivariate joint dependencies—an integration we believe to be novel in the multivariate anomaly detection(AD) literature. This aligns with earlier observations by Salinas et al., who noted that modeling correlations among variables is critical in multivariate AD. Methods that assume independence across time series fail when inter-variable correlations are critical. In anomaly detection, subtle but coordinated deviations across nodes can signal issues even if individual nodes appear normal. While prior works have acknowledged this need, to the best of our knowledge, ours is the first to bring together theoretical and architectural components in a cohesive framework with a simple self-supervised contrastive learning objective.
> > >
> > > Our experiments further illustrate this distinction: both our multivariate and copula-based models outperform baselines in Table 1. Synthetic data evaluations in Appendix E show that when marginal and joint distributions both change, the copula-based model offers a clear advantage, as they are insignificant to the changes in the marginal distribution.
> > >
> > > For latency analysis in Section C, we calculate the Average Detection Delay (ADD) using "ground truth" and observe that ADD decreases steadily during training and converges to zero for all window lengths for most datasets. In purely sample‐wise detection, there is no notion of an “event interval” to measure delay against, so ADD cannot be computed meaningfully.
> > >
> > > Regarding concept drift, we note that the five standard benchmarks used in our study do not exhibit substantial drift. To illustrate our method’s potential in such settings, we introduce artificial drift into the SWAT dataset. However, modeling AD under concept drift is not the main focus of this work, and we believe a full investigation would require access to real-world datasets exhibiting significant temporal distribution shifts—a direction we plan to pursue in future work.
> > >
> > > We hope this response clarifies our contributions and respectfully addresses your concerns. We welcome any further questions.

---

> > > > ### Author Response · Authors · 2025-07-31
> > > >
> > > > Dear Reviewer,
> > > > We have revised the final paragraph of the introduction to specifically highlight recent advances in multivariate anomaly detection that build on Transformer architectures by modifying them to capture spatio-temporal dependencies and clarify how our approach differs from these methods. The performance of these models—evaluated using precision, recall, F1-score, AUC, and accuracy—can be found in Table 1. Please let us know if there is anything further we can clarify.

---

> > > > > ### Comment · Reviewer_zeTd · 2025-08-03
> > > > >
> > > > > Thank you for the added clarifications. While the new citations emphasise why cross-variable dependency matters, they still do not provide the empirical evidence I asked for—namely, (i) published results showing a plain Transformer failing on that aspect, or (ii) a simple toy experiment illustrating the gap. Without such evidence, the motivation remains only partially supported. If you wish to soften the claim to “capturing latent dependencies is important,” that is fine, but note it also weakens the urgency for introducing a new algorithm to solve an already-acknowledged problem.
> > > > >
> > > > > That said, the other reviewers do not flag this as critical, so I will defer to the review board on the final decision. I appreciate the novel use of copulas and would be pleased to see the paper appear in TMLR if the committee is satisfied.
> > > > >
> > > > > Regarding Appendix C: the statement that ADD tends to zero is puzzling. In standard quickest-detection metrics, an ADD of zero implies an alarm immediately at event onset, which typically comes at the cost of a very high false-alarm rate. Because ADD trades off against false alarms, could you clarify whether the threshold or evaluation protocol ensures that this “zero delay” does not simply reflect a trigger-happy detector in normal periods?

---

> > > > > > ### Author Response · Authors · 2025-08-04
> > > > > >
> > > > > > Dear Reviewer, We appreciate your valuable comments again.
> > > > > >
> > > > > > We fully acknowledge the existing work on multivariate time series anomaly detection using Transformers, graphs, and other various approaches. Our intention in this paper is to contribute a new perspective to this ongoing research by focusing on a core challenge: capturing joint dependencies across variables in real-world sensor-driven cyber-physical systems.
> > > > > >
> > > > > > As Salinas et al. (NeurIPS 2019) aptly note: “Assuming independence makes such methods unsuited for applications in which the correlations between time series play an important role. In anomaly detection, observing several nodes deviating from their expected behavior can be cause for alarm even if no single node exhibits clear signs of anomalous behavior.” We believe this observation captures a critical yet underexplored aspect of anomaly detection in multivariate settings.
> > > > > >
> > > > > > Rather than emphasizing specific modeling frameworks like Transformers or Copulas, our work centers on this joint dependency modeling challenge. This focus is also reflected in the paper’s title. In order to achieve it, our method combines a Transformer (as a state-of-the-art sequence encoder) with multivariate likelihood modeling (to capture correlation), and copula theory (to model joint dependencies independently of marginals), providing a principled and adaptable framework for realistic multivariate anomaly detection.
> > > > > >
> > > > > > Furthermore, we believe that modeling in the latent space can naturally extend the solution to high-dimensional settings. In fact, the principle of "concentration of distance" (Beyer et al., 1999) states that as the dimensionality d increases, the relative contrast D between a query point's nearest and farthest neighbors approaches zero. This implies that in high-dimensional spaces, the ability to distinguish between close and distant points deteriorates significantly, reducing the discriminative power of distance-based methods.
> > > > > >
> > > > > > In response to your suggestion, we understand the value of including a plain Transformer baseline for completeness. We are currently preparing an updated set of results that includes this baseline alongside Jeong et al. (AnomalyBERT, 2023), Tuli et al. (TranAD, 2022), and Xu et al. (Anomaly Transformer, covering all key metrics such as ADD and false alarm trade-offs. we will be uploading the revised version of the paper soon.
> > > > > >
> > > > > > Thank you again!

---

> > > > > > ### Author Response · Authors · 2025-08-13
> > > > > >
> > > > > > Dear Reviewer,
> > > > > >
> > > > > > To address your comments, we have added the results of a Plain Transformer both in Table 1 (for real-world datasets and Figure 3) and in the appendix (for a synthetic toy dataset). The Plain Transformer is the standard architecture from Vaswani et al., with an additional variant incorporating the relative positional bias proposed in AnomalyBERT to capture spatial relationships. For a fair comparison, we kept the data generation process and window-wise labeling identical across models.
> > > > > >
> > > > > > From these results, we observe two noteworthy points:
> > > > > >
> > > > > > The Plain Transformer performs well on the WADI dataset, which exhibits weak or unclear correlation patterns among features—precisely where our proposed model underperforms.
> > > > > >
> > > > > > In Appendix E, Case 3 (weak joint dependency), the Plain Transformer again performs relatively well; however, its performance degrades when joint dependency is strong. This supports our hypothesis that strong spatial-domain dependencies require additional modeling strategies, such as the copula-based approach we propose.
> > > > > >
> > > > > > All code and implementation details have been made available in our GitHub repository.
> > > > > >
> > > > > > Regarding the ADD plot, our original computation was based on the validation stream, with thresholds re-selected at each epoch. This approach may bias ADD measurements, particularly when false alarms occur before the onset of a true anomaly segment. A more robust approach would be to calibrate the threshold on a normal-only slice to achieve a fixed false alarm rate, freeze it, and then compute ADD on a separate, disjoint evaluation slice.
> > > > > >
> > > > > > Finally, in reference to your comment about “published results showing a plain Transformer failing on that aspect,” the closest related work is by Jeong et al. (AnomalyBERT, 2023), which uses a Transformer equipped with relative positional bias to capture spatial relationships in multivariate data. While they generate synthetic outliers, we instead leverage a small set of real anomaly samples to synthesize similar anomaly representations. Their results, and ours, can be found in the paper for direct comparison.

---

### Review · Reviewer_h3jr · 2025-06-06

**Summary Of Contributions:**

This paper focuses on anomaly detection in multivariate time series. The aim is to identify such anomalies which may span across multiple variables, even though they may not be an anomaly with respect to any single variable. The proposed approach is to encode the time-series using a transformer, so that temporal patterns can be captured through attention mechanisms. Then a joint distribution is defined on the transformed time-series in latent space using Copulas. The idea is that a "normal" subsequence of the time-series will have higher log-likelihood than an "anomalous" subsequence, and this is enforced by contrastive learning at the time of training the parameters of the transformer and the copula. Good results are demonstrated for anomaly detection on several benchmark datasets compared to a number of state-of-the-art approaches.

**Audience:**

Yes

**Claims And Evidence:**

Yes

**Requested Changes:**

It is mentioned in the abstract that the aim is to find such anomalies which may not be detected if we focus on any individual variate of the time-series, which is why the copula approach is required. It will be good to illustrate this point with synthetic data where these kinds of anomalies are created deliberately.

While transformer seems to be a good and intuitive choice for encoding the subsequences into a latent space, it would be good to have comparisons with other models like LSTM.

It will be useful to have an illustration of the distribution of anomaly and normal subsequences in the latent space after transformation. Some t-SNE type of visualizations can make it compelling. The requirement for contrastive learning should also be highlighted this way. The dimensionality of the latent space too needs to be discussed.

Details of the benchmark datasets should be provided in a table. The performance of the proposed model is not great on the WADI dataset, but near-perfect on SWaT and other datasets. This needs to be explained with the nature of the anomalies in the different datasets. Also, for WADI dataset, recall is highest for MAD-GAN, but TranAD is highlighted.

**Strengths And Weaknesses:**

Strengths:
1) The paper proposes an intuitive approach to anomaly detection in multivariate time-series, which is based on evaluating the log-likelihood of windows of the time-series and comparing it with a threshold
2) The approach of creating latent representation using transformer and using copulas to create the joint likelihood is good and intuitive
3) Good experimental results are shown for anomaly detection on multiple benchmark datasets against other existing methods

Weaknesses:
1) The paper is light on technical innovations in ML
2) While experimental results are reported, there is less illustration on the nature of anomalies that are better detected by this approach than other approaches
3) There is also no illustration on the latent space representation of the subsequences, on the basis of which anomalies may be separated from normal subsequences

---

> ### Author Response · Authors · 2025-07-18
>
> We sincerely thank the reviewer for their thoughtful feedback, which helped us improve the clarity and depth of our work.
>
> 1. To explain this point, we construct a synthetic dataset with controlled changes in both marginal distributions and joint dependencies, and evaluate our models under various scenarios to assess their ability to capture joint structure. The results are presented in Section E of the appendix.
>
> 2. " it would be good to have comparisons with other models like LSTM" - We have compared our approach with other sequence models such as LSTMs. While the final performance is comparable, the Transformer-based model demonstrates faster convergence during training compared to the LSTM. The results are reported in section G in the appendix.
>
> 3. We show a t-SNE plot on the test data in the appendix section, Fig. 7. However, we believe that, unlike the likelihood score, t-SNE preserves neighborhood ranks, not absolute density, and may place normal and anomalous points close together despite vastly different likelihoods. Additionally, in high dimensions, pairwise distances become unreliable due to the concentration phenomenon, making t-SNE embeddings unstable. In contrast, the copula likelihood leverages full joint density, providing a more reliable measure for anomaly detection. Hence, plotting log-likelihood separation (Fig. 2)gives a clearer view of anomaly separability in the latent space as the model trains.
>
>
>    By defining a contrastive loss $\mathcal{L}(\theta,\phi)$ that rewards high log-likelihood for normal data and penalizes anomalies,
>    we can backpropagate through both the latent mapping (i.e., Transformer encoder) and the copula model
>    to learn parameters $\{\theta,\phi\}$.
>
>    We experimented with different dimensions of the latent space. Our results are presented in Section I of the appendix. The latent    space analysis reveals that projecting data into a narrow bottleneck (i.e., low-dimensional space) leads to degraded model performance. In contrast, the model performs significantly better in higher-dimensional latent spaces, such as 64 or 128 dimensions.
>
> 4. We have added dataset details and the feature description of normal and anomaly samples in section J of the appendix.

---

> > ### Comment · Reviewer_h3jr · 2025-07-26
> >
> > I thank the authors for the revision. I am generally satisfied with the response.

---

### Review · Reviewer_LyxQ · 2025-07-02

**Summary Of Contributions:**

This work examines the application of copulas to identify anomalies in time series. Using a transformer architecture, the time series is embedded and then projected on a copula. Any point from a low-density region of the distribution is considered abnormal.

**Audience:**

Yes

**Broader Impact Concerns:**

No critical concern

**Claims And Evidence:**

Yes

**Requested Changes:**

- In the abstract, the notion of a node is undefined and unclear without further context.
- While mentioned in the Appendix, the notion of copula needs to be clearly defined, with a clear description of the fitting procedure.
- While the setting appears to be the detection of abnormality in a given time series, it would be important to clearly state it as detecting abnormal time series among multiple time series could also be of interest. Furthermore, defining what constitutes an abnormality is crucial, as the assumed low-density region may not accurately reflect abnormalities depending on the chosen definition.
- $z_n(\theta)$ is undefined - is it equivalent to $f_\theta(z_n)$?
- In section 2.4, the comment on negative log-likelihood is unclear. The log-likelihood can be negative, but the aim should still be to maximize it. Currently, it reads as if the strategy ignores the likelihood associated with non-abnormalities.
- Formalise the margin constraint and explain its enforcement (in algorithm 3, it is unclear why the margin is added to the likelihood itself)
- Table 1 should contain confidence bounds to understand the significance of these results.
- Algorithm 1 is unclear on the copula definition: the normal CDF is applied twice, while $u$ should be transformed so it follows a uniform.
- In Section 3.7, the threshold selection is based on the validation set, which may misrepresent performance on a new dataset. Figure 3 should state how the threshold was chosen for precision and recall.
- How is the gradient propagated jointly? The current formulation seems to compute them separately - similar to an alternate propagation rather than a joint one.

**Strengths And Weaknesses:**

Strengths.
Explore the critical problem of time series detection.

Weaknesses.
However, the paper lacks formalization. First, both copula and abnormalities remain undefined despite being central concepts of the paper. Further, the paper assumes access to labeled abnormalities. This assumption reflects a classification setting for which multiple strategies have been proposed. Finally, some imprecisions weaken the paper (see requested changes)

---

> ### Author Response · Authors · 2025-07-18
>
> We sincerely thank the reviewer for their thoughtful feedback, which helped us improve the clarity and depth of our work.
>
> 1. "notion of a node is undefined" - In our case, a node represents spatio-temporal nodes—each corresponding to different sen-
> sensors deployed across a cyber-physical system. We have clarified it in our abstract now.
>
> 2. "the notion of copula needs to be clearly defined," - In order to model the joint dependency, we use a Gaussian and student-t multivariate likelihood model and the Gaussian and student-t copulas from the copula family. We have added the mathematical definition of copula in section 2.3.1 and described the implementation in Algo 1 and 2 separately for the two models. The training process is elaborated in Algo 3.
>
> 3. In many cyber-physical systems, changes across different time series are often interdependent. We assume that under normal operating conditions, specific patterns of dependency are expected—for example, a change in a pressure sensor is typically accompanied by a corresponding change in a temperature sensor. Anomalies or adversarial events disrupt these expected relationships, causing the system to move into low-density regions of the joint distribution.
>
> 4. $Z_n(\theta)$ represents the encoded features, which is the output of the transformer head. It is a function of the encoder parameters $\theta$. It can also be represented as $f_{\theta}(X)$ where $X$ is the original input features. We clarify the notations in Section 2.4 where we introduce the contrastive loss and also in the training algorithm (Algo 3).
>
> 5. Yes, maximizing the log‐likelihood is mathematically equivalent to minimizing its negative. We adopt the latter formulation for convenience in gradient‐based optimization, but it does not alter the underlying objective of fitting the normal‐data distribution. At inference, test samples below a chosen threshold are flagged as anomalies.  We elaborate in section 2.4.
>
> 6. We enforce the margin via a soft hinge penalty  $\min_\phi -\sum_{n=1}^N\log c_\phi(z_n)+\lambda\sum_{n=1}^N\max\{0,\tau+\delta-\log c_\phi(z_n)\}$  which encourages each $\log c_\phi(z_n)\ge\tau+\delta$ for normal samples. At test time, any sample with $\log c_\phi(z)<\tau$ is flagged as an anomaly.  We elaborate in sections 2.4 and 3.5.
>
> 7. We have added the statistical significance of the results in Table 1 now.
>
> 8. "the normal CDF is applied twice"- We begin by transforming the unknown marginals into Gaussian marginals, allowing us to validly apply the Gaussian PDF in the next step (Algo 1). Without this transformation, the random vectors may reside in an incompatible space, rendering the Gaussian PDF inapplicable. To evaluate spatial joint dependencies, we employ two distinct approaches: multivariate likelihood (Algorithm 1) and copulas (Algorithm 2). In case of Copulas, we need to apply the probability integral transform to convert the latent features to uniform marginals(step 2 in Algo 2) before applying the copula density.
>
> 9. Regarding "threshold selection," we assume the reviewer is referring to out-of-distribution (OOD) generalization on data from a different domain. In our setup, however, the train, validation, and test samples are drawn from the same dataset, with the test data representing an unseen future time period. OOD generalization is beyond the current scope of this paper. We have now referenced our threshold selection method (Section 3.5) in Figures 3 and 4 for clarity.
>
> 10. We optimize the single joint loss  $\mathcal L(\theta,\phi)=\frac{1}{N}\sum_{i=1}^N\ell^{(i)}(\theta,\phi),$
> and in one backpropagation pass compute both $\nabla_\theta\mathcal L$ and $\nabla_\phi\mathcal L$, updating $\theta$ and $\phi$ simultaneously. This implements true joint, not alternating, propagation. We elaborate the steps in Algo 3.

---

> > ### Comment · Reviewer_LyxQ · 2025-08-01
> >
> > Thank you for your detailed response and the clarifications provided. I believe these strengthen the paper overall.
> >
> > 1. I’m concerned that the current phrasing—“sensors deployed across a cyber-physical system”—may be difficult to grasp without full context. A simpler formulation, such as emphasizing the correlation between time series, might be more accessible to a broader audience.
> > 2, 3, 4, 5, 6, 7, 8, 10. Thank you for clarifying these points; I think your revisions improve the overall flow and readability of the paper.
> > 9. Apologies for the confusion, my comment referred to the threshold on the predictions used for classification, which is used in computing the evaluation metrics.
> >
> > However, my central concern remains: the paper assumes access to labeled abnormalities, which effectively places it in a classification setting. In this context, should the paper not compare its approach directly to traditional classification models? It would help clarify the contribution and practical relevance if this point was addressed more explicitly.

---

> > > ### Author Response · Authors · 2025-08-01
> > >
> > > Dear Reviewer, thank you again for your suggestions. We will be happy to address them.
> > >
> > > To address the first point, we understand that your suggestion is to keep the scope more generic in the abstract and not refer to the application so early. We have revised the abstract accordingly.
> > >
> > > We discuss the threshold selection in Section 3.4. Basically, we gather all log-likelihoods from the validation set and
> > > systematically scan a range of possible thresholds, from the minimum to the maximum observed score. We then select the threshold that maximizes the F1. The code for this section can be found in our GitHub repository for both the multivariate and copula models.
> > >
> > > We assume a small pool of real anomaly representations (≈10 %) and then generate additional anomaly sample representations via simple augmentations. In this context, the prior work spans fully unsupervised methods—TranAD (Tuli et al., ’22) and Anomaly Transformer (Jiehui et al., ’21)—and self-supervised schemes AnomalyBERT (Jeong et al., ’23). Our approach most closely follows AnomalyBert: Jeong et al. introduce a data degradation scheme for self-supervised learning by generating four synthetic outlier types and replacing input segments to train the model. By contrast, we employ contrastive learning in the representation space: positive and negative pairs—drawn from normal–normal and normal–anomaly examples—serve as pseudo-labels, and the network is trained to pull similar representations together while pushing dissimilar ones apart. Anchoring on a small set of real anomalies ensures our synthetic anomaly samples remain realistic. We try to elaborate on this in the final two paragraphs of the Introduction section. We believe that traditional supervised classification models such as SVMs and Random Forests are not suitable baselines for our problem. Instead, the three models we discussed above represent the most relevant and recent baselines for multivariate anomaly detection, as they are specifically designed to enhance Transformer architectures for this task of multivariate anomaly detection with unsupervised or self-supervised learning techniques.

---

### Decision · Action_Editor_ou5u · 2025-08-07

**Recommendation:** Accept with minor revision

**Additional Comments:**

One reviewer requests:

   "(i) published results showing a plain Transformer failing on that aspect, or (ii) a simple toy experiment illustrating the gap"

The following could satisfy the request:

 "In response to your suggestion, we understand the value of including a plain Transformer baseline for completeness. We are currently preparing an updated set of results that includes this baseline alongside Jeong et al. (AnomalyBERT, 2023), Tuli et al. (TranAD, 2022), and Xu et al. (Anomaly Transformer, covering all key metrics such as ADD and false alarm trade-offs. we will be uploading the revised version of the paper soon."

**Audience:**

Yes

**Audience Explanation:**

Anomaly detection in multivariate time series are generally of interest to the machine learning community.

**Claims And Evidence:**

Yes

**Claims Explanation:**

For multivariate time-series anomaly detection, the authors propose modeling joint dependency in the latent space via log density of multivariate likelihood estimation and via Copula log density.  A transformer is used for learning the latent space.  Evaluation of their methods include a comparison with 13 existing methods on 5 datasets.  Empirical results indicate that the proposed methods generally perform favorably against existing methods.  They also analyze the effects of varying various hyperparameters.

Two reviewers are satisfied with the revisions.  One reviewer is mostly satisfied with one revision request.